# Preference-CFR: Beyond Nash Equilibrium for Better Game Strategies

**Qi Ju** [1,2]   **Thomas Tellier** [3]   **Meng Sun** [1,2]   **Zhemei Fang** [1,2]   **Yunfeng Luo** [1,2]

## Abstract

Artificial intelligence (AI) has surpassed top human players in a variety of games. In imperfect information games, these achievements have primarily been driven by Counterfactual Regret Minimization (CFR) and its variants for computing Nash equilibrium. However, most existing research has focused on maximizing payoff, while largely neglecting the importance of strategic diversity and the need for varied play styles, thereby limiting AI's adaptability to different user preferences.

To address this gap, we propose Preference-CFR (Pref-CFR), a novel method that incorporates two key parameters: preference degree and vulnerability degree. These parameters enable the AI to adjust its strategic distribution within an acceptable performance loss threshold, thereby enhancing its adaptability to a wider range of strategic demands. In our experiments with Texas Hold'em, Pref-CFR successfully trained Aggressive and Loose Passive styles that not only match original CFR-based strategies in performance but also display clearly distinct behavioral patterns. Notably, for certain hand scenarios, Pref-CFR produces strategies that diverge significantly from both conventional expert heuristics and original CFR outputs, potentially offering novel insights for professional players.

## 1. Introduction

In machine learning, complex gaming problems are important benchmarks for assessing artificial intelligence (AI). Prominent games such as Chess (Campbell et al., 2002),

Go (Silver et al., 2016; 2018), StarCraft (Vinyals et al., 2019), and Texas Hold'em (Moravčík et al., 2017; Bowling et al., 2015; Brown & Sandholm, 2019b) have significantly influenced both academic research and public interest.

Traditionally, research has focused primarily on identifying the Nash Equilibrium (NE), as it guarantees that no player can improve their expected payoff by unilaterally deviating from their strategy. From the perspective of expected payoffs, this guarantee makes the NE considered the optimal solution in a game. Consequently, many studies treat a game problem as resolved once its NE is identified. However, maximizing expected payoffs is not the sole criterion for evaluating strategy quality.

First, many practical problems feature multiple NEs. In economics, understanding the diversity of NE often holds more value than simply identifying one. For example, Schelling's work on predicting specific NE, which earned him the 2005 Nobel Prize in Economics, exemplifies this significance.

Second, state-of-the-art game-playing AIs (e.g., AlphaGo and Pluribus) have already surpassed top human players. However, strategies with maximizing expected payoffs as the sole objective are typically overly rational, lack diversity, and are difficult for humans to master. For instance, Lee Sedol, a top human Go expert who competed against AlphaGo, chose to retire and said, "I can no longer enjoy the game." (Wakabayashi & Young, 2024) This highlights that competition is not the sole purpose of gameplay; games should also provide entertainment and spiritual fulfillment. Additionally, strategies that incur minor payoff losses against top AIs may still perform well against human opponents. For example, in chess, professional players are highly familiar with common openings recommended by AI, often resulting in drawn games. To overcome this, top players deliberately choose unconventional strategies to lead the game into unexplored scenarios, leveraging their superior skills to gain an advantage.

To the best of our knowledge, algorithms for incomplete-information games have not considered the importance of identifying diverse NEs, nor have they explored how strategy ranges change when some payoff is sacrificed. To address these limitations, we propose a novel algorithm called Preference Counterfactual Regret Minimization (Pref-CFR). This algorithm introduces two parameters for strategy selec-

[1]School of Artificial Intelligence and Automation, Huazhong University of Science and Technology [2]National Key Laboratory of Science and Technology on Multispectral Information Processing [3]GTOKing. Correspondence to: Qi Ju <juqi@hust.edu.cn>, Thomas Tellier <thomas@gtoking.com>, Zhemei Fang <zmfang2018@hust.edu.cn>.

*Proceedings of the 42$^{nd}$ International Conference on Machine Learning*, Vancouver, Canada. PMLR 267, 2025. Copyright 2025 by the author(s).

tion from the perspectives of **style** and **diversity**: the degree of preference $\delta$, representing a player's inclination toward specific strategy (the style of strategies), and the degree of vulnerability $\beta$, denoting the maximum exploitability a player is willing to tolerate (the diversity of strategies). In two-player zero-sum games, setting $\delta$ ensures convergence to a NE aligned with the specified preferences. If the game has a unique NE, setting $\delta$ alone ($\beta = 0$) does not affect the strategy's convergence. Incorporating $\beta$ further enables convergence to an $\epsilon$-NE ($\epsilon \leq \beta$), significantly expanding the selection of preferred actions. The combination of $\delta$ and $\beta$ can derive a strategy that best suits the user's style at a given tolerable loss. In Texas Hold'em experiment, we successfully obtained strategies exhibiting **Aggressive** and **Loose Passive** play-styles. Results indicate these strategies exhibit significant differences from original CFR-trained strategies while maintaining comparable performance in head-to-head matches. Notably, Pref-CFR uncovers novel strategies. For example, in the Aggressive style, it sometimes raises with weak hands like **82o**——a move previously missed by both human experts and Game Theory Optimal (GTO) solvers, offering insights for professional players. Our code can be found at GitHub.

**Related Work**

The related work in this paper is structured into two interrelated categories. First, we survey the foundational algorithms for solving game equilibria, focusing on their convergence properties and limitations. Second, we explore research that transcends traditional NE and Coarse Correlated Equilibrium (CCE), examining efforts to define "better" strategies.

In the realm of normal-form games, the preeminent algorithm is Regret Minimization (RM), along with its variant, Counterfactual Regret Minimization (CFR) (Zinkevich et al., 2007), which is applied in extensive-form games. RM/CFR can converge to the NE in two-player zero-sum games and to CCE in multi-player general-sum games (Hannan, 1957). Noteworthy variants include CFR+ (Tammelin, 2014), Monte-Carlo CFR (MCCFR) (Lanctot et al., 2009), and Discounted CFR (DCFR) (Brown & Sandholm, 2019a). In particular, CFR+ and MCCFR have played a pivotal role in significant AI advancements over the past decade (Brown & Sandholm, 2019b; Bowling et al., 2015; Brown & Sandholm, 2017). Besides the RM algorithm, the Fictitious Play (FP) algorithm is another commonly-employed approach for solving games. FP was initially introduced in Brown's 1951 paper (Brown, 1951), and the treatise *The Theory of Learning in Games* (Fudenberg & Levine, 1998) further solidified previous research, establishing a standardized framework for FP. In two-player zero-sum games, FP is proven to converge to the NE. Recently, Qi et al. integrated CFR with the FP algorithm to propose the CFVFP algorithm (Ju et al., 2024).

However, a common limitation persists: these methods primarily target NE computation, overlooking the applicability and diversity of equilibria discussed earlier.

While equilibrium-solving algorithms have advanced, game theory research has also diversified into non-NE equilibrium concepts. The 2005 Nobel Prize in Economics recognized Aumann and Schelling for refining equilibrium theory: Aumann demonstrated that correlated equilibria yield fairer and more efficient outcomes than NE (Aumann, 1974), while Schelling analyzed equilibrium selection in multi-equilibrium scenarios (Schelling, 1980). Fudenberg et al. contended that stable states reached during the learning (or evolutionary) process can also be regarded as equilibrium (Fudenberg & Levine, 1998). Recently, Ganzfried introduced a novel concept called safe equilibrium, which takes into account the irrationality of opponents and enables more flexible responses to various adversaries (Ganzfried, 2023). Despite these innovations, a critical void remains: existing studies rarely address the computational methods for solving these novel equilibria. This methodological gap motivates our synthesis of algorithmic and conceptual research streams.

This paper integrates algorithmic rigor in equilibrium solving with conceptual advances in non-NE research to bridge theory and computation. We aim to define novel stylized equilibrium strategies and demonstrate how learning theory enables their precise computation, uniting theoretical innovation with algorithmic practice.

## 2. Notion and Preliminaries

### 2.1. Game Theory

#### 2.1.1. NORMAL-FORM GAME

The normal-form game is the fundamental model in game theory. Let $\mathcal{N} = \{1, 2, \ldots, i, \ldots\}$ denote the set of players, where player $i$ has a finite action set $\mathcal{A}^i$. The strategy $\sigma^i$ of player $i$ is defined as a $|\mathcal{A}^i|$-dimensional probability distribution over $\mathcal{A}^i$ (where $|\cdot|$ represents the number of elements in the set), with $\sigma^i(a')$ indicating the probability of player $i$ choosing action $a'$. Strategies can be categorized into pure strategies and mixed strategies: a pure strategy involves taking a specific action with 100% probability, while all strategies other than pure strategies are considered mixed strategies. A strategy profile $\sigma = \underset{i \in \mathcal{N}}{\times} \sigma^i$ is a collection of strategies for all players, and $\sigma^{-i} = (\sigma^1, \ldots, \sigma^{i-1}, \sigma^{i+1}, \ldots)$ refers to all strategies in $\sigma$ except for player $i$. The set of all strategy profiles is denoted as $\Sigma = \underset{i \in \mathcal{N}}{\times} \Sigma^i$. We define the finite payoff function $u^i : \Sigma \to \mathbb{R}$, where $u^i(\sigma^i, \sigma^{-i})$ represents the payoff received by player $i$ when player $i$ selects strategy $\sigma^i$ and all other players follow the strategy profile $\sigma^{-i}$. Finally, we

define $L = \max_{\sigma \in \Sigma, i \in \mathcal{N}} u^i(\sigma) - \min_{\sigma \in \Sigma, i \in \mathcal{N}} u^i(\sigma)$ as the payoff interval of the game.

### 2.1.2. EXTENSIVE-FORM GAMES

In extensive-form games, which are commonly depicted as game trees, the set of players is represented as $\mathcal{N} = \{1, 2, \dots\}$. The nodes $s$ within the game tree signify possible states, collectively forming the state set $s \in \mathcal{S}$, while leaf nodes $z \in \mathcal{Z}$ denote terminal states. For each state $s \in \mathcal{S}$, the successor edges delineate the action set $\mathcal{A}(s)$ accessible to either a player or chance events. The player function $P : \mathcal{S} \to \mathcal{N} \cup \{c\}$ specifies which entity acts at a particular state, where $c$ represents chance.

Information sets $I \in \mathcal{I}^i$ comprise collections of states that player $i$ cannot distinguish from one another. The payoff function $R : \mathcal{Z} \to \mathbb{R}^{|\mathcal{N}|}$ assigns a payoff vector for the players based on the terminal states. The behavioral strategy $\sigma^i(I) \in \mathbb{R}^{|\mathcal{A}(I)|}$ is defined as a probability distribution over each information set $I$ for all $I \in \mathcal{I}^i$. We also define $\pi_\sigma(I)$ as the probability of encountering information set $I$ when all players select actions according to the strategy profile $\sigma$.

### 2.1.3. NASH EQUILIBRIUM

The best response (BR) strategy for player $i$ in relation to the strategy profile $\sigma^{-i}$ is defined as:

$$b^i(\sigma^{-i}) = \arg\max_{a^* \in \mathcal{A}^i} u^i(a^*, \sigma^{-i}). \qquad (1)$$

The BR strategy can either be a pure strategy or a mixed strategy; yet, identifying a pure BR strategy is generally more straightforward. For the purposes of our analysis, we will assume that $b(\sigma)$ is a pure strategy. In this context, $\arg\max$ denotes the action that produces the highest payoff within a given set. If there are multiple actions that yield this maximum payoff, the strategy that appears first in lexicographic order will be selected.

In a two-player zero-sum game, the deviation incentive of player $i$ for a strategy profile $\sigma$ is defined as:

$$\epsilon^i = u^i(b^i(\sigma^{-i}), \sigma^{-i}) - u^i(\sigma), \qquad (2)$$

while the overall exploitability $\epsilon$ across all players is calculated as:

$$\epsilon = \frac{1}{|\mathcal{N}|} \sum_{i \in \mathcal{N}} \epsilon^i, \qquad (3)$$

if $\epsilon = 0$, the strategy profile $\sigma$ is a NE; otherwise, it is termed an $\epsilon$-NE. If an algorithm ensures that the exploitability of the game meets the condition $\epsilon \le CT^{-1}$ after $T$ iterations (where $C$ is a constant), then the convergence rate of this algorithm is $O(T^{-1})$. This formula can still serve as a basis for strategy convergence in multiplayer games. However, since more than one player may deviate from the strategy simultaneously, it is no longer termed exploitability but NashConv.

### 2.2. Counterfactual Regret Minimization

In normal-form games, let $\sigma_t^i$ be the strategy used by player $i$ at iteration $t$, and the regret of player $i$ for not choosing action $a \in \mathcal{A}^i$ is defined as:

$$\bar{R}_T^i(a) = \frac{1}{T} \sum_{t=1}^T u^i(a_t^i, \sigma_t^{-i}) - u^i(\sigma_t), \qquad (4)$$

the new strategy is generated as follows:

$$\sigma_{T+1}^i(a) = \begin{cases} \frac{\bar{R}_T^{i,+}(a)}{\sum_{a \in \mathcal{A}^i} \bar{R}_T^{i,+}(a)} & \text{if } \bar{R}_T^{i,+}(a') \ne \mathbf{0} \\ \frac{1}{|\mathcal{A}^i|} & \text{otherwise,} \end{cases} \qquad (5)$$

where $\bar{R}_T^{i,+}(a) = \max(\bar{R}_T^i(a), 0)$ and $\mathbf{0}$ denotes the zero vector. Since the probability of taking action $\sigma_{t+1}^i(a)$ is proportional to the regret value $\bar{R}_T^{i,+}(a)$, this strategy is called the regret matching strategy. Define $\bar{\sigma}_T^i = \frac{1}{T} \sum_{t=1}^T \sigma_t^i$ as the average strategy of player $i$. When $T \to \infty$, $\bar{\sigma}_T^i$ converges to NE in two-player zero-sum games with a convergence rate of $O(T^{-1/2})$.

In extensive-form games, we define the counterfactual value $u(I, \sigma)$ as the expected value conditioned on reaching the information set $I$ while all players adopt the strategy $\sigma$, with the exception that player $i$ plays in a way that allows reaching $I$. For every action $a \in \mathcal{A}^i(I)$, we denote $\sigma|_{I \to a}$ as the strategy profile that is identical to $\sigma$ except that player $i$ always selects action $a$ when in information set $I$. The **average** counterfactual regret defined as:

$$\bar{R}_T^i(I, a) = \frac{1}{T} \sum_{t=1}^T \pi_{\sigma_t}^{-i}(I) \left( u^i(I, \sigma_t|_{I \to a}) - u^i(I, \sigma_t) \right), \qquad (6)$$

where $\pi_{\sigma_t}^{-i}(I)$ is the probability of information set $I$ occurring given that all players (including chance, except for player $i$) choose actions according to $\sigma_t$. We define $\bar{R}_T^{i,+}(I, a) = \max(\bar{R}_T^i(I, a), 0)$. The strategy for player $i$ at time $T + 1$ is given by:

$$\sigma_{T+1}^i(I, a) = \begin{cases} \frac{\bar{R}_T^{i,+}(I,a)}{\sum_{a \in \mathcal{A}(I)} \bar{R}_T^{i,+}(I,a)} & \text{if } \bar{R}_T^{i,+}(I) \ne \mathbf{0} \\ \frac{1}{|\mathcal{A}(I)|} & \text{otherwise.} \end{cases} \qquad (7)$$

The average strategy $\bar{\sigma}_T^i(I)$ for an information set $I$ after $T$ iterations is defined as:

$$\bar{\sigma}_T^i(I) = \frac{\sum_{t=1}^T \pi_{\sigma_t}^i(I) \sigma_t^i(I)}{\sum_{t=1}^T \pi_{\sigma_t}^i(I)}. \qquad (8)$$

Ultimately, as $T \to \infty$, $\bar{\sigma}_T$ will converge to a NE.

## 3. Motivation

### 3.1. Style and Diversity of Strategies in the Game

This paper re-examines strategy selection through the lenses of "diversity" and "style", an aspect that has been relatively

unexplored in existing research. To formalize this perspective, we first define these two metrics in the context of game theory. Specifically, **diversity** refers to the size of the acceptable strategy space $\Sigma_{\text{acc}}$, while **style** quantifies the similarity between a given strategy distribution and a preferred strategy $\sigma^*$.

The definition of $\Sigma_{\text{acc}}$ is relatively straightforward. If a player is a professional competitor aiming solely to maximize their probability of winning, the acceptable strategy space should coincide with the NE strategy set, i.e., $\Sigma_{\text{acc}} = \Sigma_{\text{NE}}$. In contrast, if the game is played casually among friends, where entertainment takes precedence over optimal strategy, then the acceptable strategy space is the set of all possible strategies, $\Sigma_{\text{acc}} = \Sigma$.

In comparison, defining "style" is more intricate. In real-world settings, styles are often measured by macroscopic statistical indicators. Roughly speaking, in football, possession percentage is commonly used to characterize Tiki-taka style——when possession exceeds a certain threshold, the playing style is classified as such. However, such statistical indicators are difficult to integrate directly into game training. As game training is inherently a sparse reward problem—where payoffs are only obtained at terminal nodes—incorporating macro-style indicators (e.g., the entry rate in Texas Hold'em) would require numerous games to generate meaningful style signals. This effectively exacerbates the sparsity issue, making learning more challenging. To address this, we map style-related measurements from the macroscopic level to the decision-making level within information sets. We define style as the distance $\text{Dis}(\bar{\sigma}_T, \sigma^*)$ between the final learned strategy $\bar{\sigma}_T$ and the preferred strategy $\sigma^*$. For instance, if a ball possession rate exceeding 70% is defined as characteristic of the Tiki-taka style, and the corresponding strategy set is $\Sigma_{\text{Tiki-taka}}$ with centroid strategy $\sigma^*_{\text{Tiki-taka}}$, then the degree to which $\bar{\sigma}_T$ adheres to the Tiki-taka style can be measured by $\text{Dis}(\bar{\sigma}_T, \sigma^*_{\text{Tiki-taka}})$.

Given these definitions, we seek a final strategy $\bar{\sigma}_T$ that minimizes the distance to the preferred style while remaining within the acceptable strategy space:

$$\bar{\sigma}_T = \min_{\sigma \in \Sigma_{\text{acc}}} \text{Dis}(\sigma, \sigma^*). \quad (9)$$

The choice of $\Sigma_{\text{acc}}$ can be guided by different criteria. In this paper, we constrain it based on the worst-case performance loss relative to NE strategies, which we term the **vulnerability degree**. Specifically, we introduce a vulnerability parameter $\beta$ during training, ensuring that the final learned strategy satisfies $\bar{\sigma}_T \in \Sigma_{\epsilon-\text{NE}}$ where $\epsilon \leq \beta$.

### 3.2. Controlling the Convergence of RM Iteration

Extensive experiments (Section 5.1 provides an example) on two-player zero-sum games reveal that even when multiple equilibria exist and the RM/CFR algorithm starts from different initial strategies, it typically converges to a unique equilibrium point. Formally, we propose the following conjecture:

**Conjecture 3.1.** *In a two-player zero-sum game, if the set of Nash equilibrium $\Sigma_{NE}$ forms a convex polyhedron, then for any initial strategy $\sigma_{t=0}$, the RM iteration converges to a unique fixed point $\sigma_{RM} \in \Sigma_{NE}$.*

This conjecture suggests that modifying the initialization of RM alone is insufficient to steer convergence towards different equilibria. Instead, altering the final strategy $\bar{\sigma}_T$ requires intervention during the iterative process. While RM lacks prior research in this direction, insights can be drawn from the Generalized Weakened Fictitious Play (GWFP) algorithm, whose update rule is:

$$\bar{\sigma}_{t+1} = (1 - \alpha_{t+1})\bar{\sigma}_t + \alpha_{t+1}(b_{\epsilon_t}(\bar{\sigma}_t) + M_{t+1}), \quad (10)$$

where $\{M_t\}_{t \geq 1}$ is a sequence of perturbation terms, and $b_{\epsilon_t}(\bar{\sigma}_t)$ is a sub-BR strategy with a gap of $\epsilon_t$ from the BR strategy, that is:

$$u^i(b(\bar{\sigma}_t^{-i}), \bar{\sigma}_t^{-i}) - u^i(b_{\epsilon_t}(\bar{\sigma}_t^{-i}), \bar{\sigma}_t^{-i}) \leq \epsilon_t. \quad (11)$$

GWFP converges to the NE when the following three conditions are met. First, $\epsilon_t \to 0$ as $t \to \infty$. Second, $\alpha_t \to 0$ as $t \to \infty$ and $\sum_{t \geq 1} \alpha_t = \infty$. Finally:

$$\lim_{t \to \infty} \sup_k \left\{ \left\| \sum_{i=t}^{k-1} \alpha_{i+1} M_{i+1} \right\| : \sum_{i=t}^{k-1} \alpha_{i+1} \leq T \right\} = 0. \quad (12)$$

We first establish that RM can be interpreted as a special case of GWFP (see Appendix D). Based on this, we propose three methods to influence RM's convergence behavior: (i) Adjusting the learning rate $\alpha_t$; (ii) Modifying the exploitability parameter in $b_{\epsilon_t}(\bar{\sigma}_t)$; (iii) Introducing a perturbation sequence $M_t$.

Among these, the most effective approach is modifying $\epsilon_t$, as we have already defined $\text{Dis}(\sigma, \sigma^*)$ as the style metric. Moreover, $\epsilon_t$ directly corresponds to the vulnerability parameter $\beta$. Therefore, our subsequent improvements will all be centered around $b_{\epsilon_t}(\bar{\sigma})$.

## 4. Method

### 4.1. Preference CFR

We now introduce the Pref-CFR algorithm. For each information set, we define a preference degree $\delta(I) \in \mathbb{R}^{|\mathcal{A}(I)|}$, where $\delta(I, a) \geq 1$. Note that in our setting, $\forall a \in \mathcal{A}(I)$ $\delta(I, a) = 1$ is not allowed. The strategy for the next itera-

tion $T + 1$ is calculated as:

$$
\sigma_{T+1}^i(I, a) = \begin{cases} \frac{\delta(I,a)\bar{R}_T^{i,+}(I,a)}{\sum_{a\in\mathcal{A}(I)}\delta(I,a)\bar{R}_T^{i,+}(I,a)} & \text{if } \bar{R}_T^{i,+}(I) \neq \mathbf{0} \\ \frac{\delta(I,a)-1}{\sum_{a\in\mathcal{A}^i(I)}\delta(I,a)-1} & \text{otherwise.} \end{cases}
$$
(13)

Alternatively, we can also adopt the BR strategy for the next iteration:

$$
\sigma_{T+1}^i(I) = \arg\max_{a\in\mathcal{A}^i(I)}\delta(I,a)\bar{R}_T^i(I,a). \quad (14)
$$

In this paper, we denote Pref-CFR(RM) to indicate that the next strategy follows the regret minimization approach, as shown in Equation 13, while Pref-CFR(BR) signifies that the next strategy is based on the BR approach, as presented in Equation 14.

We prove in the Appendix B.2 that the convergence of the Pref-CFR algorithm is consistent with the original CFR. Specifically, it can converge to the NE in two-player zero-sum games and to the CCE in multi-player general-sum games. The action preference degree $\delta(a)$ is related to the preferred strategy $\sigma^*(a)$; a larger proportion of $a'$ in $\sigma^*(a')$ warrants a higher $\delta(a')$. How to define a suitable $\delta(a)$ will be discussed in the next section.

Introducing $\delta$ drives the strategy toward convergence to different-style equilibria. However, with $\delta$ alone ($\beta = 0$), the acceptable strategy set is $\Sigma_{\text{acc}} = \Sigma_{\text{NE}}$. In many cases, the adjustments are small or have almost no macroscopic difference from non-$\delta$ strategies. As highlighted in Section 1, competition is not the sole dimension of a game. To address this, we introduce the vulnerability degree $\beta(I)$. Define:

$$
\bar{B}_T^i(I, a) = \bar{R}_T^i(I, a) - \beta(I), \quad (15)
$$

the strategy for time $T + 1$ is determined as follows:

$$
\sigma_{T+1}^i(I, a) = \begin{cases} \arg\max_{a\in\mathcal{A}^i}\bar{B}_T^i(I,a) & \text{if } \bar{B}_T^{i,+}(I,a) \neq 0 \\ \frac{\delta(I,a)-1}{\sum_{a\in\mathcal{A}^i(I)}\delta(I,a)-1} & \text{otherwise,} \end{cases}
$$
(16)

where $\bar{B}_T^{i,+}(I, a) = \max\{\bar{B}_T^i(I, a), 0\}$. In Appendix B.3, we prove that in normal-form games, the vulnerability degree $\beta$ ensures the final strategy is an $\epsilon$-NE strategy with $\epsilon \leq \beta$. By integrating preference and vulnerability, we devise a strategy that not only aligns with desired stylistic requirements but also minimizes potential losses.

### 4.2. Analysis of Pref-CFR Algorithm

Pre-CFR offers a method for converging to different strategies. Nevertheless, during actual training, the settings for $\delta(I)$ and $\beta$ still require careful configuration by experts.

1. Setting larger values for $\delta(I, a)$ and $\beta(I)$ can enhance the distinctiveness of the final strategy's style. However, this may result in slower convergence speeds and

a strategy that deviates significantly from the NE, making it susceptible to astute opponents.

2. While we can adjust $\delta(I, a)$ and $\beta(I)$ at the micro level, translating macro-level style characteristics recognized by the public into parameter settings for each information set is challenging.

To address the first problem, setting appropriate parameters for $\delta(I, a)$ and $\beta(I)$ can help the final strategy converge to a reasonable interval. For the parameter $\delta(I, a)$, experiments indicate that when $\delta(I, a) \leq 5$, the convergence speed remains acceptable, significantly increasing the likelihood of the preferred action being adopted in the final output strategy. Regarding $\beta(I)$, it can be determined based on the specifics of different games and the experience of human experts. For instance, an error margin of 25 mbb/h in Texas Hold'em is typically inconspicuous.

The second problem is more complex. Fortunately, for common games like Texas Hold'em, there is ample expert knowledge to draw upon. If a player has a narrow range of hands when entering the pot, their style is classified as "tight"; conversely, a wider range is termed "loose." For definitions of these Texas Hold'em terms, please refer to Appendix A. If a player exhibits a relatively high proportion of 3-bets after entering the pot, they are labeled as "aggressive." These styles exhibit strong consistency. Thus, we can apply the same set of $\delta$ values across all information sets. For example, to make the AI more aggressive, we could increase $\delta(I, \text{Raise})$ for all raising actions across all information sets.

However, for more extensive-form games that lack extensive expert analysis, the design of $\delta(I, a)$ and $\beta(I)$ remains an area requiring further research.

## 5. Experiments

Our experiments are conducted using Kuhn poker (Kuhn, 1950), Leduc poker (Shi & Littman, 2001) as well as two-player and three-player Texas Hold'em poker. A brief introduction to the rules of these games can be found in Appendix A. The experimental codes for Kuhn poker and Leduc poker have been made publicly available on GitHub.

In our Kuhn poker experiments, we used the vanilla Pref-CFR while Texas Hold'em experiments utilized a variant of the multi-valued state technique (Brown et al., 2018). Solutions were computed in under 10 minutes with a 24-core CPU; subgames included 35k states and the full game used for leaf estimates had 25M states. In Heads-Up play, this setup achieved exploitability below 4 mBB/h in our Texas Hold'em experiments. These settings are comparable to previous experiments (Brown et al., 2018).

## 5.1. Pref-CFR Converges to Different Nash Equilibria

Before analyzing the Pref-CFR algorithm, it is crucial to first elucidate the scenarios where CFR fails to converge to distinct NEs. We will use Kuhn poker as an example. In this game, player 1 faces multiple equilibria. The equilibrium strategy for player 1 in Kuhn poker is detailed in Table 1.

| Infoset | J_ | J_PB | Q_ | Q_PB | K_ | K_PB |
|---|---|---|---|---|---|---|
| The probability of bet | $\alpha$ | 0 | 0 | $1/3+\alpha$ | $3\alpha$ | 1 |

*Table 1.* The equilibrium strategy of player 1 in Kuhn poker, where $\alpha \in [0, 1/3]$. It can be considered that player 1 has countless equilibrium strategies in this game.

The equilibrium strategy for Player 1 can be defined by the parameter $\alpha$, which corresponds to the probability of choosing to bet in the information set J_. After iterating through $10^7$ nodes, the exploitability of the strategy is reduced to below 0.001. At this point, we approximate the equilibrium strategy with $\alpha = \sigma(\text{J}_-, \text{Bet})$.

We initialize the CFR training with a randomly distributed strategy. As depicted in Figure 1, although Kuhn poker admits multiple NEs, CFR converges to a single equilibrium at $\alpha = 0.2$, irrespective of the initial strategy. This behavior is not exclusive to Kuhn poker; as Conjecture 3.1 posits, altering the initial state of RM does not affect its final convergence point. Furthermore, the strategies generated by the CFR algorithm closely resemble traditional "machine strategies," which are complex, stable, and lacking distinct characteristics. In practical gameplay, human players can readily recognize and memorize simpler strategies, such as $\alpha = 0$ and $\alpha = 1/3$. For example, when $\alpha = 0$, player 1 always selects Pass, embodying a cautious and conservative playstyle. Conversely, Pref-CFR can identify these characteristic equilibria.

The parameter configurations for these experiments are detailed in Appendix C. As shown in Figure 2, the convergence rates across all settings remain comparable to that of the original CFR. However, all variants of Pref-CFR yield equilibrium that deviate from $\alpha = 0.2$, with higher preference degree settings resulting in more pronounced deviations. Moreover, the performance of Pref-CFR(RM) is notably inferior to that of Pref-CFR(BR). Consequently, we advocate for the use of the Pref-CFR(BR) method, which was also utilized in all subsequent experiments.

Kuhn poker, a game with multiple NEs, allows PrefCFR to converge to different NEs when $\beta = 0$. In contrast, as Fig. 3 shows, in the Leduc poker experiment, with $\delta(\text{raise}) = 10, \beta = 0$ and $\delta(\text{call}) = 10, \beta = 0$, both eventually converge like the original CFR. Only by introducing $\beta$ can the strategy deviate stably from the standard one, sug-

gesting Leduc poker may have a single NE. As analyzed in Appendix B.2, setting $\beta = 0$ implies $\Sigma_{\text{acc}} = \Sigma_{\text{NE}}$, so different $\delta$ values lead to convergence to the unique NE.

Users then face a trade-off: sacrifice some utility (compared to the NE strategy) to diversify the strategy. Fig. 4 reveals that a larger $\beta$ increases the Raise probability in the final strategy, making it more aggressive, but at the expense of higher exploitability. It is also noteworthy that in the Leduc poker experiments, the difference between setting $\delta(\text{raise}) = 10$ and $\delta(\text{raise}) = 5$ is negligible. Therefore, we set $\delta(a) = 5$ in subsequent experiments.

## 5.2. Pref-CFR Converges to Different Styles in Texas Hold'em

The traditional CFR algorithm does not account for goals beyond expected payoffs. In Texas Hold'em, two significant "style demands" have consistently emerged, yet the previous CFR algorithm was unable to address them.

1. **Aggressive strategy** A prominent feature of this strategy is its higher probability of raising. For some amateur players with an abundance of chips, sensitivity to chip loss is minimal. Their primary motivation for participating in Texas Hold'em is not necessarily to win more chips but rather to seek novel experiences or enhance the atmosphere at social gatherings. As a result, these players often look forward to engaging in games with larger pot amounts. In training, we set $\delta(I, raise) = 5$ and $\beta = 0.05$ at the first decision node of player 1.

2. **Loose passive strategy** A significant characteristic of this strategy is that the probability of folding is lower than that of a typical strategy. This phenomenon arises from the difficulty many players with a small number of chips experience when it comes to folding. After investing a considerable amount of chips during the flop or turn stages, these players often feel uncomfortable quitting the game without knowing their opponent's hole cards. The discomfort is especially pronounced when they suspect they have been successfully bluffed. By minimizing folds, this strategy provides psychological comfort for these players, helping them avoid the frustration of being outplayed by an opponent's bluff. In training, we set $\delta(I, call) = 5$ and $\beta = 0.05$ at the first decision node of Player 1.

Before analyzing style variations among AI models, we first compare their performance. Our goal is to develop AIs with distinct play styles without sacrificing significant payoffs. Given the challenge of evaluating strategy effectiveness in large-scale games, we use head-to-head matches to assess different algorithms. As shown in Table 2, performance

Kuhn Poker

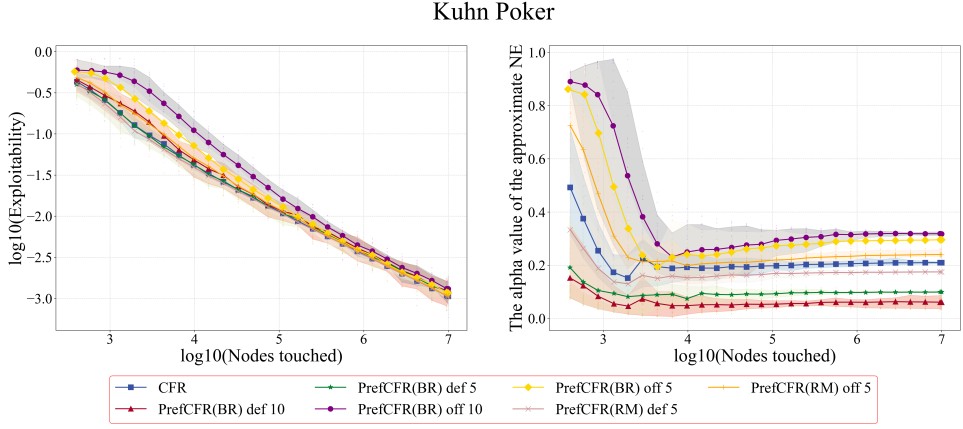

*Figure 1.* Convergence rate of CFR in Kuhn poker (left) and fluctuation of $\alpha$ in CFR algorithm iterations (right). Thirty experiments were performed for each setting, and the shaded area indicates the 90% confidence interval of these trials (the settings remain unchanged in subsequent experiments). It can be seen that regardless of the initial strategy, all CFR iterations converge to $\alpha = 0.2$.

Kuhn Poker

*Figure 2.* Convergence rate of CFR/Pref-CFR in Kuhn poker (left) and the fluctuation of the $\alpha$ value during CFR/Pref-CFR iterations (right). It is evident that the Pref-CFR algorithm can still converge to equilibrium, with a convergence speed comparable to that of the original CFR. Additionally, the right figure clearly shows that in Pref-CFR, strategies converge to results deviating from the NE, and these deviations become larger with higher values of $\beta$.

| Unit: mBB/h, 90% confidence interval: ± 1.0 | | Big Blind (Player 2) | | |
| --- | --- | --- | --- | --- |
| | | Aggressive | Normal | Loose-passive |
| Small Blind (Player 1) | Aggressive | 68.6 | 58.7 | 67.6 |
| | Normal | 80.2 | 73.2 | 79.8 |
| | Loose-Passive | 74.7 | 66.9 | 73.8 |

*Table 2.* The battle results between different AIs in two-player Texas Hold'em. In Texas Hold'em, mBB/h represents the thousandth of a big blind won or lost per hand, used to accurately measure a player's profit or loss for each individual hand.

differences across styles remain minimal in both two-player and three-player games, staying within 10mBB/h. Prior research suggests that a training error below 1mBB/h indicates convergence to NE (Bowling et al., 2015). For reference, Pluribus achieved a 32mBB/h win rate against top human players (Brown & Sandholm, 2019b). Thus, the observed 10mBB/h difference suggests these AIs are of comparable skill levels.

Our experiments clearly demonstrate how the algorithm influences the final strategies in the first row of Figure 5. In standard CFR training for two-player Texas Hold'em, the average preflop strategy distribution is [5.4%, 52.7%, 42.0%, 0.0%, 0.0%] for folding, calling, raising to 2, raising to 3, and going all-in, respectively. In contrast, the loose AI's strategy distribution shifts to [0.3%, 73.3%, 21.7%, 4.7%, 0.0%], while the aggressive AI shows [4.2%, 64.8%, 9.8%, 21.1%, 0.0%]. As expected, the loose AI exhibits a much lower folding probability, dropping from 5.4% to 0.3%—a 94.4% decrease. The differences between aggressive and standard strategies are also pronounced, with the aggressive AI raising to 3 at a 21.1% rate compared to 0% in the standard strategy.

We believe these AI strategies provide valuable insights for human players. For example, traditional human ex-

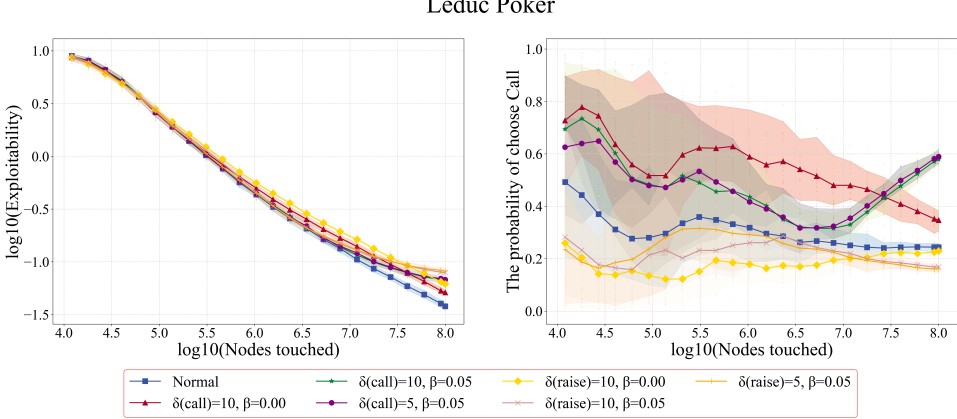

*Figure 3.* Convergence rates of ES-MCCFR/Pref-ES-MCCFR in Leduc poker (left) and the fluctuations in the probability of choosing Call during ES-MCCFR/Pref-ES-MCCFR iterations (right). This figure shows that in Leduc poker, strategies will converge to different equilibria only when $\beta > 0$ is set.

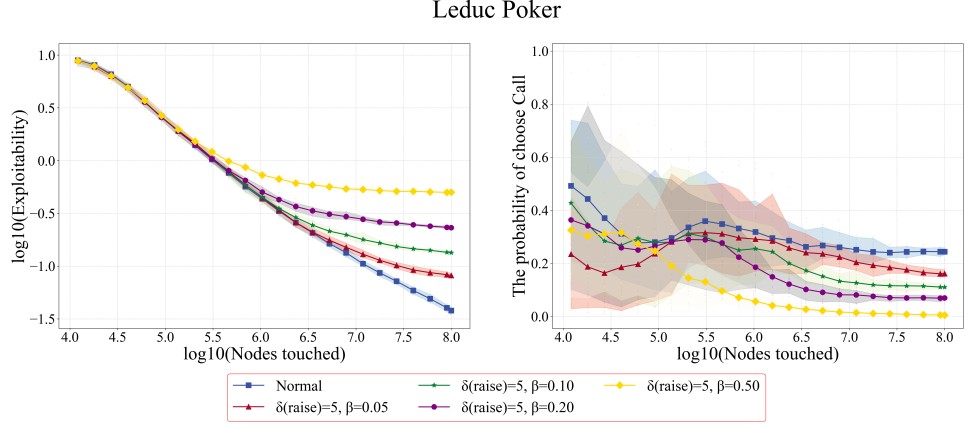

*Figure 4.* Convergence rates of ES-MCCFR/Pref-ES-MCCFR in Leduc poker (left) and the fluctuations in the probability of choosing Call during ES-MCCFR/Pref-ES-MCCFR iterations (right). Obviously, the larger $\beta$ is, the higher the probability of choosing to Call and the more obvious the strategy style is.

perts and GTO-based AIs typically fold weak hand combinations like 82o and 72o, considered poor due to low straight potential, unsuitedness, and low card ranks. In contrast, our loose-passive AI opts to call almost 100% of the time, while the aggressive AI may even raise with these hands. Such stylized strategies cleverly adjust hand distributions to obscure hand strength, akin to "hiding a leaf in the forest." The loose-passive AI's higher calling rates make it difficult for opponents to distinguish between a weak-hand bluff or a strong-hand slow-play. Similarly, the aggressive AI increases raising frequency across all hands, potentially challenging conventional Texas Hold'em tactics. While loose-passive play was historically deemed suboptimal, our results show it can be a viable strategy when balanced across all hand combinations, offering a fresh tactical perspective. The results of three-player AI matches,

| Unit: mBB/h, 90% confidence interval: ± 0.9 | | Small Blind (Player 2) & Big Blind (Player 3) | | |
| --- | --- | --- | --- | --- |
| | | Aggressive | Normal | Loose-passive |
| Button (Player 1) | Aggressive | 247.2 | 251.0 | 259.1 |
| | Normal | 256.3 | 252.2 | 260.0 |
| | Loose-Passive | 253.7 | 252.4 | 253.9 |

*Table 3.* The battle results between different AIs in three-player Texas Hold'em.

displayed in Table 3 and the second row of Figure 5, show that our algorithm achieves even more pronounced effects in three-player scenarios. In standard three-player training, the average strategy distribution is [61.4%, 0.0%, 38.6%, 0.0%]. This example illustrates why the loose-passive style was previously considered ineffective: according to GTO calculations, it is typically not recommended. However, our

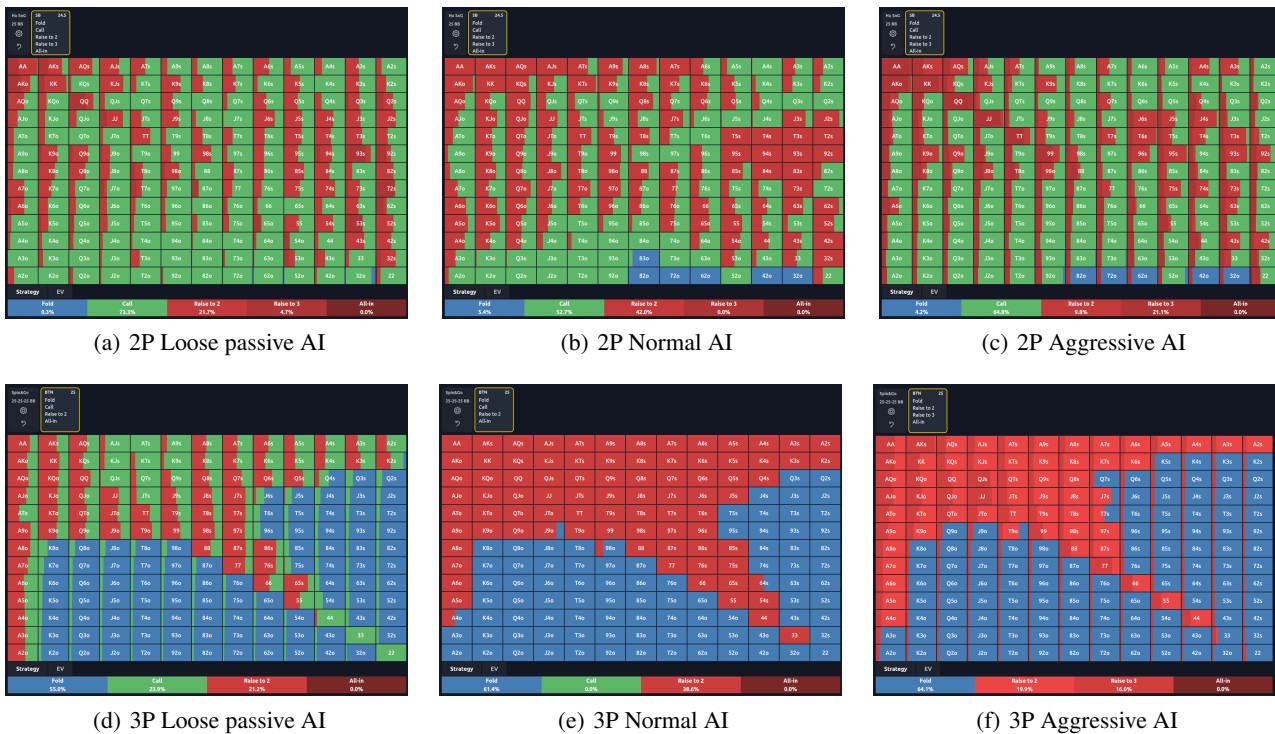

(a) 2P Loose passive AI  (b) 2P Normal AI  (c) 2P Aggressive AI

(d) 3P Loose passive AI  (e) 3P Normal AI  (f) 3P Aggressive AI

*Figure 5.* Strategy display for Texas Hold'em. In the top left corner of each image, the current player's information and available actions are displayed. The central area showcases the strategies for different hand combinations at this stage. In Texas Hold'em poker, there are 13 ranks across 4 suits, with no distinction in value between suits, resulting in 169 unique hand combinations. These are represented in a 13×13 matrix, where the lower left displays offsuit hands and the upper right shows suited hands. Each matrix element's color indicates the strategic choice for the corresponding hand: blue for folding, green for calling, red shades for raising (with deeper red shades indicating higher raises), and black-red for going all-in. The bottom row provides an overview of the average strategies across all hands, allowing for a visual understanding of the overall strategy distribution.

experiments reveal that the Loose-Passive strategy not only avoids any loss in earnings but actually gains an advantage of 0.2mBB/h compared to the Normal strategy, while significantly increasing the calling probability from 0.0% to 23.9%. We speculate that the added complexity of three-player poker compared to two-player games accentuates the impact of style changes.

## 6. Conclusion and Prospect

The Pref-CFR algorithm proposed in this study addresses a key limitation of traditional CFR by enabling the discovery of diverse equilibria. Our experiments demonstrate successful training of Texas Hold'em AIs with distinct strategic profiles. For example, in three-player games, the Aggressive AI increased its 3-bet probability from 0.0% to 16.0% with only a 10mBB/h performance decrement, while the Loose AI raised its calling rate from 0.0% to 23.9%. Notably, the algorithm maintains training efficiency and supports real-time strategic guidance for human players.

However, the current framework requires manual calibration

of preference degrees for actions in each information set, restricting its adaptability to varied player styles. Given the game's complexity, human users face challenges in implementing context-dependent strategies. To enhance practical utility, we propose automating the translation of user-specified metrics (e.g., betting frequencies or pot entry rates) into information-set-specific preference weights, streamlining the customization workflow. Additionally, extending this framework to interdisciplinary domains—such as behavioral economics or market dynamics—presents an intriguing avenue for bridging game-theoretic algorithms with real-world decision systems.

## Impact Statement

This paper presents work whose goal is to advance the field of Machine Learning. There are many potential societal consequences of our work, none of which we feel must be specifically highlighted here.

## Acknowledgements

This work was supported by the National Natural Science Foundation of China (Grant No. 62103158).

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

# A. Game introduction

## A.1. Kuhn poker

Kuhn poker is a simple poker game. Here is a detailed rules introduction to it:

- **Card composition**: Kuhn poker uses only three cards: J, Q, and K. Each player can only get one card at a time.

- **Action sequence**: Usually, two players participate in the game. At the beginning of each round, each player places a blind bet first. Then each player will randomly receive a card. Subsequently, players take actions in turn. The action options are **Bet** and **Pass**. If one player bets, the other player can choose to call(Bet) or fold(Pass).

- **Winning determination**: Finally, if a player calls, both players show their cards and compare the sizes. The player with the larger card wins and takes the chips in the pot. In Kuhn poker, K > Q > J.

Kuhn poker is a classic model in game theory research. Due to its simple rules and limited card types and action choices, it is convenient for mathematical analysis and theoretical derivation.

## A.2. Leduc poker

Leduc poker is a simplified poker game that extends Kuhn poker with additional complexity, making it a key model for game theory research. Here is a detailed introduction to its rules:

- **Card composition**: Leduc poker uses six cards from two suits (e.g., spades and hearts), specifically J, Q, K of each suit. Each player is dealt one hole card, and one community card is placed face up.

- **Action sequence**: The game involves two players and two betting rounds:
    1. **First round**: Players post blinds, then receive hole cards. Actions include **Bet** or **Check** (pass). If a player bets, the opponent can **Call** or **Fold**.
    2. **Second round**: A community card is dealt, and players act again. Actions remain Bet, Check, Call, or Fold, with betting limits typically set to standardize stakes.

- **Winning determination**: If both players call by the end of the second round, they reveal their hole cards. Hand strength is determined by:
    1. **Pair**: Hole card and community card of the same rank (e.g., J-J) is the strongest.
    2. **High card**: If neither pair nor flush, compare the higher card (K ¿ Q ¿ J); if tied, spades suit prevails over hearts.

Leduc poker bridges the gap between simple and complex games, featuring two betting rounds and community cards that introduce strategic depth. Its structured complexity—more intricate than Kuhn poker but less complex than Texas Hold'em—makes it ideal for testing algorithms in extensive-form games and studying equilibrium strategies in partial-information scenarios. Our experiments used a 12-card (6-pair) setup.

## A.3. Texas Hold'em

Texas Hold'em is a poker game that has a large player population and extensive influence globally. In terms of tournaments, numerous international Texas Hold'em competitions draw significant attention. High prize money attracts many top players to participate. Socially, it is a popular activity for people to enhance communication during leisure gatherings. The following is an introduction to some basic Texas Hold'em poker terminology. For more content, you can refer to Wikipedia.

- **Preflop**: The stage before the community cards are dealt. Decisions are made based on hands, position, etc., like raising, calling or folding.

- **Flop**: The first three community cards dealt. Evaluate competitiveness combined with hands and decide strategies.

- **Turn**: The fourth community card after the flop.

- **River**: The last community card. Decides the final decision.

- **Big blind (BB)**: The big blind is a forced bet made by one of the players before the cards are dealt. It is typically twice the size of the small blind.

- **Small blind (SB)**: The small blind is also a forced bet made by a player before the cards are dealt.

- **Button**: The button indicates which player is the dealer for that hand. The player on the button has certain advantages in terms of position and play order.

- **mBB/h**: In Texas Hold'em, mBB/h represents the thousandth of a big blind won or lost per hand, used to accurately measure a player's profit or loss for each individual hand.

- **Loose**: Play many hands with a low entry criterion. Often participate even with weak hands. High risk but may win with weak hands against strong ones.

- **Aggressive**: Actively bet or raise to put pressure on opponents and control the game to win the pot. The hand may not be strong.

- **3 Bet**: Raise again after someone has raised. Often indicates a strong hand or wanting to pressure opponents and increase the pot.

- **All-in**: Bet all chips. Due to confidence in the hand or having few chips and wanting to pressure.

- **Call**: Follow by betting the same amount of chips as the opponent. Want to continue to see the cards and compete.

- **Fold**: Give up the hand and not participate in the pot. Due to weak hands or high risk from large bets by opponents.

- **Bluff**: When having a weak hand, make large bets and let opponents think the hand is strong so they fold to win the pot. High risk.

# B. Proof of Convergence of Pref-CFR

## B.1. Blackwell Approachability Game

**Definition B.1.** *A Blackwell approachability game in normal-form two-player games can be described as a tuple $(\Sigma, u, S^1, S^2)$, where $\Sigma$ is a strategy profile set, $u$ is the payoff function, and $S^i = \mathbb{R}_{\leq 0}^{|\mathcal{A}^i|}$ is a closed convex target cone. The Player $i$'s regret vector of the strategy profile $\sigma$ is $R^i(\sigma) \in \mathbb{R}^{|\mathcal{A}^i|}$, for each component $R^i(\sigma, a_x) = u^i\left(a_x, \sigma^{-i}\right) - u^i(\sigma)$, $a_x \in \mathcal{A}^i$ the average regret vector for players $i$ to take actions at $T$ time $a$ is $\bar{R}_T^i$*

$$\bar{R}_T^i = \frac{1}{T} \sum_{t=1}^{T} R^i(\sigma_t), \tag{17}$$

*at each time $t$, the two players interact in this order:*

- *Player 1 chooses a strategy $\sigma_t^1 \in \Sigma^1$;*

- *Player 2 chooses an action $\sigma_t^2 \in \Sigma^2$ , which can depend adversarially on all the $\sigma_t$ output so far;*

- *Player 1 gets the vector value payoff $R^1(\sigma_t) \in \mathbb{R}^{|\mathcal{A}^1|}$.*

*The goal of Player 1 is to select actions $\sigma_1^1, \sigma_2^1, \ldots \in \Sigma^1$ such that no matter what actions $\sigma_1^2, \sigma_2^2, \ldots \in \Sigma^2$ played by Player 2, the average payoff vector converges to the target set $S^1$.*

$$\min_{\hat{s} \in S^1} \left\| \hat{s} - \bar{R}_T^1 \right\|_2 \to 0 \quad as \quad T \to \infty. \tag{18}$$

Before explaining how to choose the action $\sigma_t$ to ensure this goal achieve, we first need to define the forceable half-space:

**Definition B.2.** *Let $\mathcal{H} \subseteq \mathbb{R}^d$ as half-space, that is, for some $\boldsymbol{a} \in \mathbb{R}^d$, $b \in \mathbb{R}$, $\mathcal{H} = \left\{\boldsymbol{x} \in \mathbb{R}^d : \boldsymbol{a}^\top \boldsymbol{x} \le b\right\}$. In Blackwell approachability games, the halfspace $\mathcal{H}$ is said to be forceable if there exists a strategy $\sigma^{i*} \in \Sigma^i$ of Player $i$ that guarantees that the regret vector $R^i(\sigma)$ is in $\mathcal{H}$ no matter the strategy played by Player $-i$, such that*

$$R^i\left(\sigma^{i*}, \hat{\sigma}^{-i}\right) \in \mathcal{H} \quad \forall \hat{\sigma}^{-i} \in \Sigma^{-i}, \tag{19}$$

*and $\sigma^{i*}$ is forcing action for $\mathcal{H}$.*

Blackwell's approachability theorem states the following.

**Theorem B.3.** *Goal 18 can be attained if and only if every halfspace $\mathcal{H}_t \supseteq S$ is forceable.*

The relationship between Blackwell approachability and no-regret learning is:

**Theorem B.4.** *Any strategy (algorithm) that achieves Blackwell approachability can be converted into an algorithm that achieves no-regret, and vice versa (Abernethy et al., 2011).*

If the algorithm achieves Blackwell approachability, the average strategy $\bar{\sigma}_T^i$ will converge to equilibrium with $T \to \infty$. The rate of convergence is $\epsilon_T^i \le \bar{R}_T^i \le L\sqrt{|\mathcal{A}^i|}/\sqrt{T}$.

## B.2. Pref-CFR Achieves Blackwell Approachability

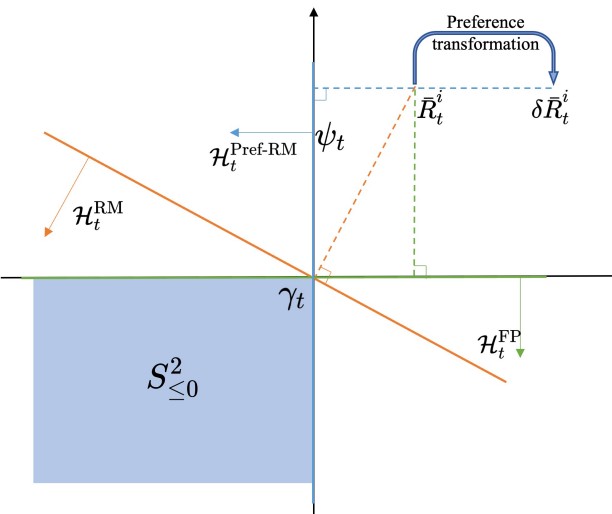

*Figure 6.* The differences in the selected forcing actions and forcing half-spaces of RM, FP and Pref-RM in the two-dimensional plane.

The proof concept for the Pref-CFR algorithm is as follows:

If an algorithm guarantees that the regret value converges to zero in normal-form games, applying this algorithm to two-player zero-sum games will converge to the NE and to the CCE in multi-player general-sum games. Furthermore, Theorem B.4 indicates that any algorithm satisfying Blackwell approachability is equivalent to a no-regret algorithm. Additionally, as shown in paper (Zinkevich et al., 2007), if an algorithm ensures that regret converges to zero in normal-form games, then its application in extensive-form games leads to the convergence of counterfactual regret to zero (i.e., behavioral strategy convergence to NE in two-player zero-sum games). Thus, to demonstrate that Pref-CFR converges to NE in two-player zero-sum extensive-form games, we need only establish that it satisfies Blackwell approachability.

In this proof, we primarily show that Pref-CFR(BR) meets the criteria for Blackwell approachability, while Pref-CFR(RM) can derive similar results through a relatively straightforward conversion.

Let $\bar{R}_t^{i,*}$ represent any component of the vector $\bar{R}_t^i$ such that $\bar{R}_t^{i,*} > 0$. Next, we identify the point $\boldsymbol{\psi}_t = \bar{R}_t^i - \bar{R}_t^{i,*} \in \mathbb{R}^{|\mathcal{A}^i|}$ on the axis (noting that this point is not necessarily located on the surface of the target cone $S^i$). We can define the normal vector as $\frac{\bar{R}_t^i - \boldsymbol{\psi}_t}{|\bar{R}_t^i - \boldsymbol{\psi}_t|}$, which allows us to determine the half-space defined by this normal vector and point $\boldsymbol{\psi}_t$:

$$\mathcal{H}_t^{\mathrm{P}} = \left\{\boldsymbol{z} \in \mathbb{R}^{|\mathcal{A}^{-i}|} : (\bar{R}_t^i - \boldsymbol{\psi}_t)^\top \boldsymbol{z} \le (\bar{R}_t^i - \boldsymbol{\psi}_t)^\top \boldsymbol{\psi}_t\right\}. \tag{20}$$

Since $\bar{R}_t^i - \psi_t = \bar{R}_t^{i,*}$ and $(\bar{R}_t^i - \psi_t)^\top \psi_t = 0$, we can simplify this to:

$$\mathcal{H}_t^{\mathrm{P}} = \left\{ z \in \mathbb{R}^{|\mathcal{A}^{-i}|} : \left\langle \bar{R}_t^{i,*}, z \right\rangle \leq 0 \right\}, \tag{21}$$

for any point $s' \in S^i$ there is $\left\langle \bar{R}_t^{i,*}, s' \right\rangle \leq 0$. Then we need to find the forcing action that matches $\mathcal{H}_t^{\mathrm{P}}$. According to Definition B.2, we need to find a $\sigma_{t+1}^{i*} \in \Sigma^i$ that achieves $R^i \left( \sigma_{t+1}^{i*}, \hat{\sigma}_{t+1}^{-i} \right) \in \mathcal{H}_t^{i,\mathrm{P}}$ for any $\hat{\sigma}_{t+1}^{-i} \in \Sigma^{-i}$. For simplicity, let $\ell = [u^i \left( a_1, \sigma^{-i} \right), \dots]^\top \in \mathbb{R}^{|\mathcal{A}^i|}$, we rewrite the regret vector as $R^i \left( \sigma_{t+1}^{i*}, \hat{\sigma}_{t+1}^{-i} \right) = \ell - \left\langle \ell, \sigma_{t+1}^{i*} \right\rangle \mathbf{1}$, we are looking for a $\sigma_{t+1}^{i*} \in \Sigma^i$ such that:

$$
\begin{aligned}
&R^i \left( \sigma_{t+1}^{i*}, \hat{\sigma}_{t+1}^{-i} \right) \in \mathcal{H}_t^{\mathrm{P}} \\
\Longleftrightarrow\quad &\left\langle \bar{R}_t^{i,*}, \ell - \left\langle \ell, \sigma_{t+1}^{i*} \right\rangle \mathbf{1} \right\rangle \leq 0 \\
\Longleftrightarrow\quad &\left\langle \bar{R}_t^{i,*}, \ell \right\rangle - \left\langle \ell, \sigma_{t+1}^{i*} \right\rangle \left\langle \bar{R}_t^{i,*}, \mathbf{1} \right\rangle \leq 0 \\
\Longleftrightarrow\quad &\left\langle \bar{R}_t^{i,*}, \ell \right\rangle - \left\langle \ell, \sigma_{t+1}^{i*} \right\rangle \left\| \bar{R}_t^{i,*} \right\|_1 \leq 0 \\
\Longleftrightarrow\quad &\left\langle \ell, \frac{\bar{R}_t^{i,*}}{\left\| \bar{R}_t^{i,*} \right\|_1} \right\rangle - \left\langle \ell, \sigma_{t+1}^{i*} \right\rangle \leq 0 \\
\Longleftrightarrow\quad &\left\langle \ell, \frac{\left[ \bar{R}_t^i \right]^*}{\left\| \left[ \bar{R}_t^i \right]^* \right\|_1} - \sigma_{t+1}^{i*} \right\rangle \leq 0.
\end{aligned}
\tag{22}
$$

Therefore, the strategy $\sigma_{t+1}^{i*} = \frac{\bar{R}_t^{i,*}}{\left\| \bar{R}_t^{i,*} \right\|_1}$ can guarantee $\mathcal{H}_{t+1}^{\mathrm{P}}$ to be forceable half-space. Figure 6 intuitively shows the relationship between these points, half-spaces and the target set in a two-dimensional plane.

It should be noted that the half-space $\mathcal{H}_{t+1}^{\mathrm{P}}$ can only prove that the distance from the average regret value $\bar{R}_t^i$ to the point $\psi_t$ converges to $\mathbf{0}$, but $\psi_t$ is not necessarily on the target cone $S^i$. So it cannot be shown that $\bar{R}_t^i$ will converge to $S^i$. The projection of $\bar{R}_t^i$ to the target cone $S^i$ is point $\gamma_t^i = \left[ \bar{R}_t^i \right]^-$, $\bar{R}_t^{i,-} = \min\{0, \bar{R}_t^i\}$. We need to keep the distance from the $\bar{R}_t^i$ to point $\psi_t$ and point $\gamma_t^i$ within a certain range, and preference degree $\delta(a)$ just plays this role.

Define $a^{\mathrm{BR}} = \arg\max_{a \in \mathcal{A}^i} \bar{R}_t^i(a)$, $a^P = \arg\max_{a \in \mathcal{A}^i} \delta(a) \bar{R}_t^i(a)$. The distance from the average regret value $\bar{R}_t^i$ to $\gamma_t^i$ is $\|\bar{R}_t^{i,+}\|_2$, and the distance from $\bar{R}_t^i$ to the point $\psi_t$ is $\delta^i(a^P) \bar{R}_t^i(a^P)$. Define $\mathrm{dist}(x, y)$ as the distance between points $x$ and $y$.

$$\mathrm{dist}(\bar{R}_t^i, \gamma_t) = \|\bar{R}_t^{i,+}\|_2 \leq \sqrt{|\mathcal{A}^i|} \bar{R}_t^{i,+}(a^{\mathrm{BR}}), \tag{23}$$

at the same time,

$$\mathrm{dist}(\bar{R}_t^i, \psi_t) = \delta^i(a^P) \bar{R}_t^{i,+}(a^P) \geq \delta^i(a^{\mathrm{BR}}) \bar{R}_t^{i,+}(a^{\mathrm{BR}}), \tag{24}$$

since $\delta^i(a^{\mathrm{BR}}) \geq 1$, so:

$$\mathrm{dist}(\bar{R}_t^i, \gamma_t) \leq \frac{\sqrt{|\mathcal{A}^i|} \delta^i(a^P)}{\delta^i(a^{\mathrm{BR}})} \mathrm{dist}(\bar{R}_t^i, \psi_t). \tag{25}$$

Define $\delta^* = \max_{a \in \mathcal{A}^i}(\delta^i(a))$. Therefore, from Blackwell's approachability, we know that $\mathrm{dist}(\bar{R}_t^i, \psi_t) \leq \frac{L\sqrt{|\mathcal{A}^i|}\delta^*}{\sqrt{t}}$. Finally, we get:

$$\mathrm{dist}(\bar{R}_t^i, \gamma_t) = \frac{L|\mathcal{A}^i|\delta^*}{\sqrt{t}}. \tag{26}$$

The convergence speed will linearly slow down as $\delta^*$ increases. As long as $\delta^*$ is set within a reasonable range, it will not significantly reduce the convergence speed of Pref-CFR.

### B.3. Vulnerability CFR Will Converge to an $\epsilon$-NE

The proof for the Vulnerability CFR algorithm is more straightforward. In the original scenario, the historical strategy $\bar{\sigma}_t^i$ converging to a NE is equivalent to the regret $\bar{R}_t^i$ converging to the convex target cone $S^i = \mathbb{R}_{\leq 0}^{|\mathcal{A}^i|}$. This holds true because,

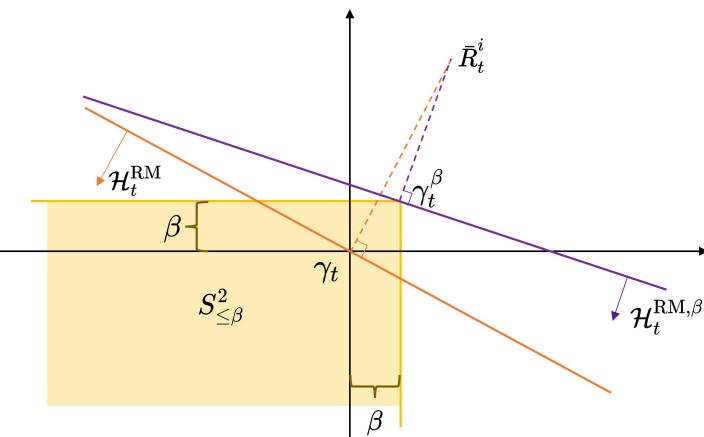

*Figure 7.* The differences in the selected forcing actions and forcing half-spaces of RM and $\beta$ vulnerability RM in the two-dimensional plane.

given the opponent's strategy $\bar{\sigma}_t^{-i}$, player $i$ identifies a strategy $\bar{\sigma}_t^i$ such that regardless of which pure strategy player $i$ selects, their payoff will not increase (since $\bar{R}_t^i(a) \leq 0$ for all $a \in \mathcal{A}^i$ as $t \to \infty$), thus satisfying the conditions for NE.

However, what if our target cone is not $S^i = \mathbb{R}_{\leq 0}^{|\mathcal{A}^i|}$, but instead $S^i = \mathbb{R}_{\leq \beta}^{|\mathcal{A}^i|}$ for some $\beta \geq 0$? This means that the regret of any pure strategy will not exceed $\beta$, indicating that we have identified a $\beta$-NE.

In the iteration, we only need to adjust the translation. The original projection point is $\gamma_t = \bar{R}_t^{i,-}$, and the distance that needs to be reduced is given by:

$$\text{dist}(\bar{R}_t^i, \gamma_t) = \|\bar{R}_t^i - \gamma_t\|_2 = \|\bar{R}_t^{i,+}\|_2. \tag{27}$$

Now, the projection point becomes $\gamma_t^\beta = \bar{R}_t^{i,-\beta}$, where $\bar{R}_t^{i,-\beta} = \min\{\bar{R}_t^i, \beta\}$. At this point, the distance that needs to be reduced is:

$$\text{dist}(\bar{R}_t^i, \gamma_t^\beta) = \|\bar{R}_t^i - \gamma_t^\beta\|_2 = \|\bar{R}_t^{i,+} - \beta\|_2. \tag{28}$$

Thus, in the CFR iteration, we simply need to replace $\bar{R}_t^i(a)$ with $\bar{B}_t^i(a) = \bar{R}_t^i(a) - \beta$ to ensure that the final strategy converges to a $\beta$-NE. Figure 7 intuitively illustrates the relationship between these points, the half-spaces, and the target set in a two-dimensional plane.

Strictly speaking, in many of our subsequent experiments, both Pref-CFR and Vulnerability CFR are used simultaneously. However, Pref-CFR is the more critical component of the algorithm. Using Vulnerability CFR alone does not produce meaningful results. Therefore, even though Vulnerability CFR is utilized, it will not be mentioned in the name of the algorithm.

## C. Kuhn Poker Experiment Setup

The settings in the experiment are as follows:

1. Pref-CFR(**BR**) with $\delta(J\_,\text{Bet}), \delta(Q\_,\text{Bet}), \delta(K\_,\text{Bet}) = \mathbf{10}$.

2. Pref-CFR(**BR**) with $\delta(J\_,\text{Bet}), \delta(Q\_,\text{Bet}), \delta(K\_,\text{Bet}) = \mathbf{5}$.

3. Pref-CFR(**RM**) with $\delta(J\_,\text{Bet}), \delta(Q\_,\text{Bet}), \delta(K\_,\text{Bet}) = \mathbf{5}$.

4. Pref-CFR(**RM**) with $\delta(J\_,\text{Pass}), \delta(Q\_,\text{Pass}), \delta(K\_,\text{Pass}) = \mathbf{5}$.

5. Pref-CFR(**BR**) with $\delta(J\_,\text{Pass}), \delta(Q\_,\text{Pass}), \delta(K\_,\text{Pass}) = \mathbf{5}$.

6. Pref-CFR(**BR**) with $\delta(J\_,\text{Pass}), \delta(Q\_,\text{Pass}), \delta(K\_,\text{Pass}) = \mathbf{10}$.

In Kuhn poker, there are 12 information sets. For all actions in all information sets except for those on the information sets specified here, $\delta(I, a) = 1$.

## D. The RM Algorithm is a GWFP Process

The final strategy of RM is:

$$\bar{\sigma}_T = \frac{1}{T} \sum_{t=1}^{T} \sigma_{t,\text{RM}}, \tag{29}$$

where

$$\sigma_{t,\text{RM}} = \begin{cases} \frac{\bar{R}_T^{i,+}(a)}{\sum_{a \in \mathcal{A}^i} \bar{R}_T^{i,+}(a)} & \text{if } \bar{R}_T^{i,+}(a') \neq \mathbf{0} \\ \frac{1}{|\mathcal{A}^i|} & \text{otherwise,} \end{cases} \tag{30}$$

Compared with Formula 10, in the GWFP process, it is equivalent to $\alpha_t = 1/t$, $M_t = \mathbf{0}$, and $b_{\epsilon_t} = \sigma_{t,\text{RM}}$. Therefore, in a two-player zero-sum game, we only need to prove

$$\lim_{T \to \infty} \epsilon_T = 0, \tag{31}$$

where

$$\epsilon_T = u^i(b^i(\bar{\sigma}_T^{-i}), \bar{\sigma}_T^{-i}) - u^i(\sigma_{T,\text{RM}}^i, \bar{\sigma}_T^{-i}), \tag{32}$$

to show that the RM algorithm is a GWFP process.

First, recall that in the FP process, the essence is to find a BR strategy to the historical strategy.

$$\bar{\sigma}_{t+1} = \frac{t}{t+1}\bar{\sigma}_t + \frac{1}{t+1}b(\bar{\sigma}_t). \tag{33}$$

In a two-player zero-sum normal-from game, this means performing a matrix multiplication. Let the pay-off matrix of the game be $U \in \mathbb{R}^{|\mathcal{A}^1| \times |\mathcal{A}^2|}$. Then, to find $b(\bar{\sigma}_t)$, we need to calculate

$$b^1(\bar{\sigma}_t^2) = \arg\max_{a \in \mathcal{A}^1} U\bar{\sigma}_t^2. \tag{34}$$

Here, we start from the perspective of player 1, and the calculation method for player 2 is symmetric. Specifically:

$$\begin{aligned} b^1(\bar{\sigma}_t^2) &= \arg\max_{a \in \mathcal{A}^1} U\bar{\sigma}_t^2 = \frac{1}{t} \arg\max_{a \in A} U \sum_{k=1}^{t} \sigma_k^2 \\ &= \frac{1}{t} \arg\max_{a \in \mathcal{A}^1} \sum_{k=1}^{t} U\sigma_k^2. \end{aligned} \tag{35}$$

Define $q_t^i(a) = u^i(a, \sigma_t^{-i})$ and $Q_T^i = \sum_{t=1}^{T} q_t$, then

$$b^i(\bar{\sigma}_t^{-i}) = \arg\max_{a \in \mathcal{A}^i} Q_t^i. \tag{36}$$

Define $\bar{Q}_T^i = \frac{1}{T}Q_T^i$, and the value obtained is:

$$u^i(b(\bar{\sigma}_T^{-i}), \bar{\sigma}_T^{-2}) = \bar{Q}_T^i(b(\bar{\sigma}_T^{-i})), \tag{37}$$

the value obtained by the RM strategy is:

$$u^i(\sigma_{T,\text{RM}}^i, \bar{\sigma}_T^{-i}) = \sum_{a \in \mathcal{A}^i} \sigma_{T,\text{RM}}^i(a)\bar{Q}_T^i(a). \tag{38}$$

When $\bar{R}_T^{i,+}(a') = \mathbf{0}$, it means that all $\bar{Q}_T^i(a)$ are equal, and naturally:

$$u^i(b(\bar{\sigma}_T^{-i}), \bar{\sigma}_T^{-i}) - u^i(\sigma_{T,\text{RM}}^i, \bar{\sigma}_T^{-i}) = 0. \tag{39}$$

When $\bar{R}_T^{i,+}(a') \neq \mathbf{0}$, in the RM process, define $\bar{V}_T^i = \sum_{t=1}^{T} u^i(\sigma_t)$. The regret of each action can be rewritten as:

$$\bar{R}_T^i(a) = \bar{Q}_T^i(a) - \bar{V}_T^i, \tag{40}$$

Formula 32 can be rewritten as:

$$\begin{aligned}
\epsilon_T^i &= u^i(b(\bar{\sigma}_T^{-i}), \bar{\sigma}_T^{-i}) - u^i(\sigma_{T,\text{RM}}^i, \bar{\sigma}_T^{-i}) \\
&= \bar{Q}_T^i(b(\bar{\sigma}_T^{-i})) - \sum_{a \in \mathcal{A}^i \text{ and } \bar{R}_T^{i,+}(a) > 0} \sigma_{T,\text{RM}}^i(a) \bar{Q}_T^i(a) \\
&= \sum_{a \in \mathcal{A}^i \text{ and } \bar{R}_T^{i,+}(a) > 0} \sigma_{T,\text{RM}}(a) \left( \bar{Q}_T^i(b(\bar{\sigma}_T^{-i})) - \bar{Q}_T^i(a) \right) \\
&= \sum_{a \in \mathcal{A}^i \text{ and } \bar{R}_T^{i,+}(a) > 0} \sigma_{T,\text{RM}}(a) \left( \bar{R}_T^i(b(\bar{\sigma}_T^{-i})) - \bar{R}_T^i(a) \right).
\end{aligned} \tag{41}$$

Since $\bar{R}_T^{i,+}(a) > 0$, so:

$$0 < \bar{R}_T^i(a) \le \bar{R}_T^i(b(\bar{\sigma}_t^{-i})), \tag{42}$$

in a two-player zero-sum game, $\lim_{T \to \infty} \max_{a \in \mathcal{A}^i} \bar{R}_T^i(a) = 0$. We have:

$$\epsilon_t^i = \sum_{a \in \mathcal{A}^i \text{ and } \bar{R}_T^{i,+}(a) > 0} \sigma_{T,\text{RM}}(a) \left( \bar{R}_T^i(b(\bar{\sigma}_T^{-i})) - \bar{R}_T^i(a) \right) = 0, \tag{43}$$

holds for all $i \in \mathcal{N}$. Therefore, RM satisfies all the conditions of GWFP, and RM is a GWFP process. Q.E.D.

