# OpenReview forum: "Preference-CFR: Beyond Nash Equilibrium for Better Game Strategies"
_ICML.cc/2025/Conference — ICML 2025 poster_

### Official Review · Reviewer_XPTK · 2025-03-13

**Overall Recommendation:** 3

**Summary:**

The paper introduces Preference-CFR (Pref-CFR), an extension of Counterfactual Regret Minimization (CFR) aimed at generating diverse strategies in extensive-form games. Traditional CFR-based approaches focus on solving for Nash Equilibria (NE), which prioritize optimality under worst-case conditions but lack strategic diversity. Pref-CFR introduces preference degree and vulnerability degree to control playstyle and allowable exploitability, respectively. The method enables the training of AI agents with distinct styles, such as Aggressive and Loose Passive strategies, in Texas Hold’em. Experimental results show that Pref-CFR achieves comparable performance to standard CFR-trained strategies while demonstrating diverse playstyles.

**Claims And Evidence:**

Yes

**Essential References Not Discussed:**

No.

**Experimental Designs Or Analyses:**

Yes

**Methods And Evaluation Criteria:**

Yes

**Other Comments Or Suggestions:**

See strengths and weaknesses

**Other Strengths And Weaknesses:**

Strengths
- Novel conceptual contribution: Introducing preference-based parameters into CFR is an interesting extension.
- Empirical validation: The method performs well in Texas Hold’em, an established benchmark.
- Theoretical soundness: The algorithm has theoretical proof on convergence.

Weaknesses
- Heuristic tuning of $\delta$ and $\beta$ requires manual selection. The sensitivity of parameter selection in large-scale game is not discussed.

**Questions For Authors:**

How sensitive are the results to the heuristic tuning of  $\delta$ and $\beta$ , particularly in large-scale games?

**Relation To Broader Scientific Literature:**

Traditional CFR and its extensions (CFR+, DCFR, MCCFR).
Correlated Equilibria and other refinements beyond NE.
Opponent-exploiting AI (e.g., safe equilibrium models), which could serve as an additional baseline.

**Theoretical Claims:**

I didn't check the proof of theory section in the appendix in detail.

---

> ### Author Rebuttal · Authors · 2025-04-01
>
> Thank you for your feedback. We summarize some of the questions and answer them. If there are still questions or new questions arise, please feel free to discuss further.
>
> ---
>
> ## How sensitive are the results to the heuristic tuning of $δ$ and $β$, particularly in large-scale games?
>
> Conducting experiments in large-scale games is highly costly, and comprehensive analysis of these two parameters would require significant time and hardware resources. However, we have successfully migrated the PrefCFR method to OpenSpiel, a game learning framework developed by DeepMind. By employing a larger game instance of Leduc Poker, we can conduct detailed analysis of these parameters' effects. The results can be viewed at: https://drive.google.com/file/d/1F-oNqLm_cVqBNHvDYUf5dtQ_Rk1SdMkH/view?usp=drive_link
>
> Experiments show that the introduction of the preference parameter alone has little effect on the convergence of the game. Even if it is set to 10, the convergence speed is only reduced but not significantly. When fragility is introduced, the convergence speed in the early stage of training is basically the same as that of the original CFR algorithm, and only slows down when it approaches $β$.
>
> ---
>
> ## Heuristic tuning of $δ$ and $β$ requires manual selection.
>
> Regarding manual selection of $δ$ and $β$: Manual parameter selection is currently required, but in the context of Texas Hold'em poker - where player styles have distinct categorizations and sufficient expert knowledge exists - this manual selection does not create a bottleneck for the current algorithm implementation. We plan to further investigate this aspect in future research.

---

### Official Review · Reviewer_CeZM · 2025-03-13

**Overall Recommendation:** 4

**Summary:**

This paper tackles a key limitation of standard CFR-based methods. Typically, these methods converge to a single Nash equilibrium (NE) and struggle to accommodate risk preferences or different playing styles. To address this, the authors propose Preference-CFR (Pref-CFR), which introduces two key parameters at each decision point: a preference degree, $\delta(I,a)$, that amplifies or suppresses specific actions, and a vulnerability degree, $\beta(I)$, which controls how much exploitability is allowed, guiding the solution toward a $\epsilon$-NE instead of a strict NE.

Using Blackwell approachability, the authors show that Pref-CFR maintains low regret, ensuring strong near-NE performance. In Kuhn poker experiments, the method steers strategy toward different equilibria by adjusting $\delta$, whereas standard CFR always converges to the same equilibrium. In larger-scale tests on two and three player Texas Hold'em, Pref-CFR effectively shapes playstyles, such as increasing aggression or reducing folds, resulting in distinctive, human-like strategies like "aggressive" or "loose-passive". These adjustments maintain high performance with only a slight trade-off in exploitability.

**Claims And Evidence:**

Most of the paper's claims are reasonably supported, but there are places where the evidence could be more thorough. The authors assert that Pref-CFR can not only converge to different equilibria but also preserve near-optimal performance. They demonstrate examples of altered strategies in Kuhn poker and Texas Hold'em, along with head-to-head matches measuring exploitability. If we look at the claim that increasing the preference parameters $\delta$ or the vulnerability parameters $\beta$ "does not significantly slow down convergence," the supporting data come primarily from small-scale Kuhn experiments and selected subgame analyses in Texas Hold'em. These examples, while informative, do not fully guarantee scalability or robustness in larger or more complex scenarios, especially when very large $\delta$ or $\beta$ values are applied across numerous information sets.

While the experiments demonstrate a few distinct playstyles, such as "aggressive" and "loose-passive," they don't rigorously test whether the method can handle a large number of style parameters simultaneously or adapt to more nuanced, context-specific preferences. It would have been useful to see broader experiments systematically exploring a wider range of preference settings, such as analyzing how performance changes when $\delta$ is adjusted incrementally across all betting rounds. Without these deeper evaluations, it's fair to question whether the method truly scales to diverse playstyles as seamlessly as the paper suggests. The claims aren't entirely unsupported, but more extensive and systematic testing would have made the evidence far more compelling.

**Essential References Not Discussed:**

Since this paper is grounded in CFR models, it would be beneficial to include a more comprehensive discussion of related works within this domain. While the current related work section primarily provides background information on existing approaches, it lacks a direct comparison between these methods and the proposed approach. A stronger comparative analysis would help contextualize the contributions of this paper by highlighting how it improves upon prior work. Specifically, discussing the strengths and limitations of other CFR-based methods and demonstrating where your approach offers advantages would make the contribution clearer. This would not only strengthen the justification for your method but also provide a more complete picture of the research landscape

**Experimental Designs Or Analyses:**

The experiments in the paper are generally appropriate for illustrating the potential of the proposed method, yet there are a few points:

* The authors test on Kuhn poker (a very small game) and on subgame approximations of Texas Hold'em (two and three player). This does show that their algorithm can scale beyond toy scenarios, but the evaluation is limited in how comprehensively it inspects all stages of a large game. By focusing on subgame methods, we do not necessarily see full-tree exploitability or style changes across every betting round.
* The experiments tend to vary only a handful of preference degrees $(\delta)$ or vulnerability degrees $(\beta)$ in fairly simple ways (e.g., drastically increasing raise frequency or almost never folding). It is plausible that moderate or more nuanced adjustments could produce different equilibrium behaviors. This keeps the experiments intuitive, yet it leaves open questions about how the method behaves when many parameters are tweaked in more subtle ways.

Overall, the experimental designs serve their main purpose, showing that Pref-CFR can shift an equilibrium strategy's style with minimal performance loss.

**Methods And Evaluation Criteria:**

The choice of poker, Kuhn and Texas Hold'em, as a testing ground does make sense, given that poker has a long history in computational game theory. The proposed approach, Pref-CFR, specifically targets scenarios where one needs flexible or style-driven equilibria rather than a single exploitability-minimizing solution, and poker is a natural fit for demonstrating that kind of variety.

* While Kuhn poker is a classic toy example and works nicely to illustrate theoretical proofs, it's extremely small in scale and doesn't necessarily expose all the complications that arise in real-world poker settings. The paper then jumps to two and three player Texas Hold'em, which is certainly more complex, but the presentation of final strategies (e.g., aggregated preflop charts) and small-sample matchups may not fully convey whether the style modifications maintain robustness across the entire decision tree. More rigorous exploitability analyses, or thorough cross-play among multiple different stylized agents, might have better established how well the approach generalizes.

* While the evaluation criteria of measuring head-to-head performance in terms of mBB/h is standard in poker, there's some risk in concluding broad "style" success based primarily on changes to preflop charts. One could argue that style adjustments should manifest consistently across multiple betting rounds, and the current paper only provides a snapshot of the final strategy. So, even if the overall benchmarks make sense, the completeness of the evaluation might be limited when it comes to verifying that the agent truly embodies a different style (as opposed to just shifting its action frequencies slightly at the earliest decision points).

In summary, the methods and evaluation criteria make sense given the application, but they could be more thorough in probing deeper behaviors and broader ranges of scenarios.

**Other Comments Or Suggestions:**

* The introduction section would benefit from a more technical explanation of key concepts and terminology to better introduce the research area to the reader.
* The technical background in Section 2 is well-presented and effectively enhances the clarity of the paper.

**Other Strengths And Weaknesses:**

Strengths:
* Novel Extension to CFR: The introduction of preference and vulnerability parameters is a straightforward yet effective modification of the CFR framework, allowing for different equilibria based on user-defined stylistic or risk constraints.
* Practical Demonstration: Experiments on both Kuhn poker and Texas Hold'em games convincingly show how adjusting the parameters leads to distinct, human-like playing styles (aggressive, loose-passive) with minimal performance degradation.

Weaknesses:
* While the experiments are illustrative, they focus primarily on a few stylized adjustments and measure changes mostly at preflop or initial decision points. Postflop actions in full-scale Texas Hold'em and extended multi-street analysis are not exhaustively demonstrated.
* The related work section primarily provides background information but lacks a direct comparison with similar CFR-based methods. A more comprehensive discussion of related works and a stronger comparative analysis would better highlight the contributions of this paper. Addressing the strengths and limitations of existing approaches while emphasizing the advantages of the proposed method would clarify its impact and strengthen its justification.

**Questions For Authors:**

1. Your experiments mostly highlight changes in preflop or early-round decision-making. Did you conduct any deeper analyses on later betting rounds to confirm that style-related shifts remain consistent throughout the entire game tree? If so, do you observe similarly clear style distinctions in postflop play?
2. While it is shown that extreme values can push the final strategy toward distinct equilibria, have you observed any surprising instability when $\delta$ or $\beta$ become very large (for example, extremely aggressive or extremely passive settings)? Does the algorithm remain stable, or do you see any oscillations or convergence slowdowns in such scenarios?

**Relation To Broader Scientific Literature:**

see next section

**Theoretical Claims:**

The proofs in the paper appear consistent with known results on no-regret learning and Blackwell approachability. The authors' main technical claim is that, by introducing preference parameters $\delta(I,a)$ and vulnerability $\beta(I)$ in the update steps, they can still guarantee convergence to a (possibly $\beta$-relaxed) Nash equilibrium. Their proofs hinge on two well-known building blocks: (1) standard regret-minimization arguments, and (2) Blackwell approachability ensuring that the average regret vector converges.

---

> ### Author Rebuttal · Authors · 2025-04-01
>
> Thank you for your feedback. We will first answer your questions and comments in order, and then summarize and answer some questions from previous sessions. If there are still questions or new questions arise, please feel free to discuss further.
>
> ---
>
> ## Questions 1
>
> Our algorithm adopts a similar idea to Pluribus, that is, only the pre-flop strategy is saved, and the subsequent strategy is obtained by real-time search, so we only show the per-flop results.
>
> ---
>
> ## Questions 2
>
> Conducting such an experiment directly in large-scale problems is extremely costly. Therefore, we carried out more experiments in Leduc. The experimental charts can be found at: https://drive.google.com/file/d/1F-oNqLm_cVqBNHvDYUf5dtQ_Rk1SdMkH/view?usp=drive_link
>
> No unexpected instability was found in the results, which is in line with theoretical expectations, which also proves the robustness of our algorithm.
>
> In Figure 1, we doubled the value of $\delta$ (from 5 to 10). It can be seen that the improvement in the stylistic characteristics of the final result is negligible. Thus, we have every reason to believe that a larger $\delta$ is unnecessary.
>
> In Figure 2, we set different values of $\beta$. It can be observed that the influence of different $\beta$ values on the results is fully consistent with the theoretical expectations:
> 1. The $\epsilon$ of the final strategy will not exceed the set $\beta$;
> 2. A larger $\beta$ can lead to a more pronounced stylistic strategy.
>
> Tips: In this game, there are only two actions, namely "raise" and "call", and $\sigma(\text{raise}) + \sigma(\text{call}) = 1$.
>
> ---
>
> ## More extensive preference experimentation
>
> This seems unnecessary because a strategy cannot have two styles simultaneously. For example, at the beginning, we set $\delta$ as an aggressive strategy (with a relatively large $\delta(raise)$), and then change it to a conservative strategy in the middle of the training process (reducing the value of $\delta(raise)$), which seems meaningless. However, this suggestion has inspired us. When training strategies of different styles, we don't have to start the training from a completely random strategy. Instead, we can first train a "Normal" strategy, and then use this "Normal" strategy as a warm start. Subsequently, strategies of different styles can continue to be trained based on this "Normal" strategy. We will try this approach in our subsequent work.
>
> ---
>
> ## Discussing the strengths and limitations of other CFR-based methods
>
> Indeed, we lack the comparison with previous methods, and there is also a lack of technical introductions of key concepts and terms in the introduction stage. We will add this part of content in the subsequent versions.

---

### Official Review · Reviewer_BR1q · 2025-03-14

**Overall Recommendation:** 3

**Summary:**

This paper introduces **Preference Counterfactual Regret Minimization (Pref-CFR)**, an extension of **Counterfactual Regret Minimization (CFR)**, designed to incorporate **strategy diversity** and **playstyle customization** in game AI. While standard CFR focuses on computing **Nash Equilibrium (NE)**, the authors argue that optimality alone is insufficient for practical applications requiring diverse strategies.

### **Main Contributions** ###

Pref-CFR introduces **Preference and Vulnerability Degrees** to adjust strategy tendencies, enabling AI to exhibit different playstyles while maintaining competitiveness.

### **Experimental Results** ###

Applied to **Texas Hold’em**, Pref-CFR successfully learns **Aggressive** and **Loose Passive** playstyles, performing competitively with standard CFR-trained strategies while demonstrating distinct stylistic differences.

**Claims And Evidence:**

- #### **Strengths:** ####

  - **Pref-CFR retains the convergence properties of CFR** and can still reach **Nash Equilibrium (NE)** under certain conditions, as theoretically proven in **Appendix B.2**.
  - The algorithm extends to **ϵ-NE**, demonstrating mathematical soundness even with some loss tolerance.
  - Pref-CFR introduces **novel strategy discoveries**, such as **raising with weaker hands**, which may offer insights for professional players.

- #### **Weaknesses:** ####

  - While theoretically applicable to other **extensive-form games**, there is **no empirical validation** beyond **Texas Hold’em**, limiting its demonstrated generalizability.
  - The **lack of human player testing** makes it uncertain whether Pref-CFR’s strategies provide meaningful improvements for real-world play.

**Essential References Not Discussed:**

NA.

**Ethical Review Concerns:**

NA.

**Experimental Designs Or Analyses:**

- #### **Strengths:** ####

    - The experimental design effectively validates Pref-CFR’s ability to train AI with different playstyles in Texas Hold’em, demonstrating both Aggressive and Loose Passive strategies.

    - Pref-CFR’s competitiveness is assessed by comparing it with standard CFR-trained AI, showing that distinct playstyles can be achieved without significant performance loss.

    - The algorithm discovers new strategies , suggesting that Pref-CFR may offer novel insights beyond conventional CFR-based approaches.

- #### **Weaknesses:** ####

    - Limited scope: Experiments are restricted to Texas Hold’em, with no validation in other imperfect information games , limiting generalizability.

    - Lack of human player evaluation: The study does not test whether Pref-CFR-generated strategies are beneficial for real-world human players, which is crucial given its focus on strategic diversity.

    - No comparison with advanced game AIs: The paper does not benchmark Pref-CFR against state-of-the-art models like DeepStack and Pluribus, making it unclear whether it offers meaningful advantages.

    - Future research should expand experimental settings and incorporate human feedback to further validate Pref-CFR’s real-world applicability.

**Methods And Evaluation Criteria:**

- #### **Strengths:** ####

    - Pref-CFR effectively **adjusts strategic tendencies** through **preference degree and vulnerability degree**, enabling AI to learn diverse playstyles while maintaining competitiveness.
    - The **evaluation criteria are reasonable**, focusing on **strategy diversity** and **competitiveness**, which align with common practices in game AI research.
    - AI playstyle variation is **demonstrated through parameter adjustments**, providing an important metric for evaluating strategic flexibility.

- #### **Weaknesses:** ####

    - **Limited validation**: The method has only been tested in **Texas Hold’em**, while **CFR applies to a broader range of imperfect information games**. Additional experiments in **other domains (e.g., StarCraft, economic games)** are needed to establish generalizability.
    - **Lack of human player feedback**: Pref-CFR aims to improve strategies for human players, but **no real-world testing with professional players** has been conducted.
    - **No comparison with state-of-the-art game AIs**: The study does not evaluate **whether more advanced AI models can achieve similar playstyle control**, leaving Pref-CFR’s comparative advantage uncertain.

**Other Comments Or Suggestions:**

- **Clarify Scope of Applicability**: The paper primarily conducts experiments in **Texas Hold’em**, but its applicability to other **imperfect information games** remains unclear. Discussing how the method could generalize to **multi-agent strategy games, economic simulations, and other domains** would strengthen the paper's contributions.

 - **Consider Human Player Evaluation**: Since the method aims to adjust strategy styles, incorporating **feedback from professional players** would help assess its practical effectiveness in real-world gameplay.

 - **Improve Formula Notation and Explanation**: Some **formula symbols and derivations** could be further clarified, particularly in **Appendix B**, to enhance readability.

 - **Figure 3 Strategy Visualization Improvements**:
  (1) The **current strategy display for Texas Hold’em** is informative but lacks clarity and aesthetic appeal.
  (2) Consider a **more intuitive representation**, such as a cleaner color scheme or a different layout to enhance interpretability

**Other Strengths And Weaknesses:**

- #### **Weaknesses:** ####

 - **Limited experimental scope**: The experiments are currently restricted to **Texas Hold’em**, lacking validation in other imperfect information games (e.g., **StarCraft, economic games, bridge**).

 - **No comparison with more advanced game AIs**: The advantages of Pref-CFR in strategy style control and competitiveness have not been fully evaluated against state-of-the-art AI models.

 - **Lack of human player evaluation**: If Pref-CFR aims to generate strategies that align better with human playstyles, professional players should be involved in testing, and their feedback should be collected to assess its usability and impact on human players.

- **Limited practical significance**: While the generated strategies show diverse playstyles, the **real-world impact** of such stylistic variations remains unclear. What is the **practical value** of these playstyles, and how do they **contribute to improving gameplay** in real-world settings?

**Questions For Authors:**

- **Is Pref-CFR applicable to a broader range of imperfect information games?**
 The current experiments are limited to **Texas Hold’em**, but **CFR** is applicable to a wider range of games.

- **Have the strategies generated by Pref-CFR been tested by human players?**
 Is there a plan to have **professional players** test the strategies generated by Pref-CFR and collect feedback to verify their feasibility in real-world gameplay?

- **What is the practical value of having different playstyles in the generated strategies?**
 Given that the generated playstyles (e.g., Aggressive and Loose Passive) are largely theoretical, how do these stylistic variations **translate into real-world advantages**, particularly for human players? Does this diversity actually improve gameplay, or is it primarily a theoretical construct?

**Relation To Broader Scientific Literature:**

This study builds on **CFR**, which has been widely used in imperfect information games. However, traditional CFR focuses only on **optimal strategy** and lacks control over **strategy diversity**. This paper introduces **preference degree and **vulnerability degree, allowing AI to **adjust strategies across different playstyles**, filling the gap in **personalized strategy learning** and providing a new direction for research on **adaptive strategies in game AI**.

**Theoretical Claims:**

The core theoretical proofs in the paper are correct, and the mathematical derivations contain no obvious errors.

---

> ### Author Rebuttal · Authors · 2025-04-01
>
> Thank you for your detailed feedback. The strengths of the paper were perfectly understood, so we will only address weaknesses and questions below.
>
> ---
> ## Comparisons to state-of-the-art AI
> Our algorithm is not aimed at defeating top-level AI, so we didn't conduct this experiment at the start. However, competing against top AI can also prove the effectiveness of our algorithm, so we added this part of the experiment.
>
> For this paper, the algorithm and the strategy profile it produces needed to satisfy the following constraints:
> 1. Style customization, and possibly its usefulness
> 2. Beta-exploitability
> 3. Ability to work on a large-scale game
>
> Challenging a state-of-the-art AI is a third-party verification that ensures the correctness of the strategy profile produced and its lack of exploitability (2). It proves (3) even more, since it would use the full game (without bet or cards abstraction).
>
> However, it is not a direct way to evaluate the exploitability of the algorithm, which we did later in Head’s Up Texas Hold’em Poker. As expected, exploitability was under Beta. To be precise, **passive solutions were very close to the normal version of CFR (6 mBB/h vs 4 mBB/h), while aggressive versions were very close to Beta (23 mBB/h vs 25 mBB/h).** We believe that this difference is explained by the average size of the pot: in terms of %pot exploitability, results are closer.
>
> Now, to get back to the state-of-the-art AI point, **the blueprint (no real-time solving) strategy used in the paper defeated Slumbot 2019 after 100,000 hands, by 34.2 mBB/h $\pm$ 12 mBB/h**, which is a performance similar to the one achieved by Baby Tartanian8 against Slumbot 2017. Moreover, while blueprints are more exploitable in later streets due to post-flop cards’ abstraction, preflop is barely exploitable, and we only showed preflop results.
>
> Finally, you mentioned “the paper does not benchmark Pref-CFR against state-of-the-art models like DeepStack and Pluribus, making it unclear whether it offers meaningful advantages”. Pref-CFR should offer no advantage against top AIs, its goal is not to play optimally but to offer a trade-off between customization and performance, and to allow humans to understand how much wiggle room they have when deviation from Game Theory Optimal (GTO) solutions.
>
> ---
>
> ## Is Pref-CFR applicable to a broader range of imperfect information games?
>
> **In fact, the scope of application of our algorithm is consistent with CFR, which is proved in our Appendix B.2.** To avoid misunderstanding, we will emphasize this result in the main text. In addition,
>
> In addition, **We have integrated Pref-CFR into OpenSpiel**, a game-theoretic learning framework that supports over 100+ games. We successfully conducted two additional experiments in this environment. Due to character limitations, this part of the experiment can be found in the reply of reviewer CeZM. However, large-scale games require more specialized frameworks, which are often closed source, so we will limit ourselves to Texas Hold’em for now.
>
> ---
> ## Have the strategies generated by Pref-CFR been tested by human players?
>
>
>
>
>
> Hiring professional players for evaluation is extremely challenging. For example, Pluribus paid more than 50,000 dollars for professional players to appear in the experiment. The focus of our paper is to propose a method to find different styles of strategies in the game. It is too early to play against professional players. We conducted some tests among amateur players to prove that our algorithm is effective, and we will add this part in subsequent versions.
>
> ---
>
> ## What is the practical value of having different playstyles in the generated strategies？
>
> In Texas Hold'em, we analyze the practicality of our algorithm from two perspectives: **novel experience** and **winning more money**.
> If it is to give players a novel experience, our current strategy is already very recognizable. In fact, our strategy can already become a qualified companion AI. We have verified the content of this part in the games played by amateur players.
>
> However, if you want to win more money with this strategy, there are two difficulties.
> 1. The strategies are hardly memorizable: there is a mixed strategy for every combination of private cards, requires players to practice for a long time.
> 2. If a player plays a particular style in order to win money, it is often to exploit a particular opponent’s weakness, while we do it without direct purpose.
>
> For the above two problems, we can use preference degree and beta to model the opponent's strategy profile, and then find the best response that is less likely to be exploited. Here, Pref-CFR will be used for modeling. It requires sufficient domain knowledge to model such a role, as well as an algorithm that produces safe exploits and meets player-friendly characteristics. We will continue to improve our algorithm in subsequent research.

---

> > ### Comment · Reviewer_BR1q · 2025-04-06
> >
> > Thank you for the detailed response. It has addressed my main concerns.
> > I am raising my score to 3 (Weak Accept).

---

> > > ### Author Response · Authors · 2025-04-07
> > >
> > > Thanks  for your acknowledge of our work and for raising score. Thanks again  for your time and effort  in reviewing  our paper.

---

### Official Review · Reviewer_FBND · 2025-03-23

**Overall Recommendation:** 2

**Summary:**

This paper proposes a method for finding approximate equilibrium strategies in extensive-form games and claims to achieve strategies that balance “style” and “diversity” with playing strength. The authors present theoretical ideas in the context of two-player, zero-sum (2P0S) games (e.g., Kuhn poker), and also provide preliminary experiments in multiplayer Texas Hold’em. Their overall goal is to develop agents that have low exploitability while also satisfying certain preferences about how the agent plays.

**Claims And Evidence:**

- The paper claims that their method produces strategies which play according to the desired "style", while not increasing exploitability by too much. They claim this produces interesting, low-exploitability strategies in poker, whereas existing AI strategies for poker all play in the same style.

The experimental results shown in Figures 1 and 2 (Kuhn poker) are convincing in demonstrating the method’s effectiveness in a toy domain. They convincingly show that their method can result in strategies that behave very differently, and without the cost of increasing exploitability by much.
For the Texas Hold’em experiments, the evidence that the trained agents are low-exploitability is not as strong. The authors’ discussion under Table 2 attempts to convince the reader of the strength of the agents, but they are unconvincing to me. More direct exploitability evaluation on a smaller domain where exploitability can be calculated (e.g., Leduc poker, Turn/River hold'em) would strengthen the claims.

**Essential References Not Discussed:**

-

**Experimental Designs Or Analyses:**

See: Methods and Evaluation Criteria.

**Methods And Evaluation Criteria:**

The proposed method is a modification of CFR, parameterized by two parameters. The proposed evaluation criteria are Kuhn poker, 2-player Texas hold'em, and 3-player Texas hold'em.

The paper needs to be much more clear about which parts apply to the 2P0S (two-player zero sum) setting only, and which claims pertain to the general setting. Almost nowhere in the paper do the authors specify that they specifically care about 2P0S, yet much of the discussion makes sense only in the 2P0S setting (e.g. characterization of Nash equilibria in Introduction as "maximizing expected payoffs under worst-case scenarios", and the rest of the paragraph and following paragraphs) -- e.g. the CFR-based methods surely only have convergence guarantees in 2P0S settings, yet experiments are also run on a 3-player domain.

As mentioned above, there is not convincing evidence that the Texas Hold'em experiments produced low-exploitability agents. This is very difficult to do in a domain like Texas hold'em, so experiments in a more tractable domain would improve the paper.

**Other Comments Or Suggestions:**

- Figures 1 and 2 are excellent results. They are convincing evidence of the effectiveness of the paper's method (albeit in a toy domain), and also clearly communicates the aims of the research direction. The charts could use some cleanup though -- all the text/labels in the charts are too small and hard to read.

**Other Strengths And Weaknesses:**

In all, the paper is well-motivated, the research direction is especially interesting to the poker community. I did not review the proofs of the methods for convergence, but the results in Kuhn poker look very good, and the qualitative Texas hold'em results are interesting. Nevertheless, I'm hesitant to recommend this version of the paper for this conference -- the writing still needs a lot of work. Additionally, the Texas hold'em experiments don't contain enough detail.

**Questions For Authors:**

- In Introduction, "Nevertheless, strategies with winning probability as the sole objective" should probably say something like "EV" or "maximizing expected payoffs" instead of "winning probability", to be precise (for domains like chess with loss/draw/win and poker with integer-valued payoffs).
- Bottom of page 1/top of page 2: The descriptions of \beta and \delta sound a little silly to me: "style" corresponds to the degree of exploitability of the strategy, and "diversity" corresponds to closeness to a particular type of play? Neither of those sound intuitive.
- "setting only \delta" is not the most understandable -- this means setting \beta to 0, right? Why not just say that?
\epsilon = \beta? Why do we need the notation for both then? Or, should this say \epsilon \leq \beta?
- Related Work (line 94/95): "ju et al." should be capitalized?
- Should clarify that FP converges in 2P0S.
- I don't find the paragraph starting with line 100 to be super relevant -- too broad. Besides, CEs are NEs in 2P0S.
- "This paper will synthesize the research from these two aspects" -- what two aspects?
- Section 2.1.2 Extensive-Form Games: Last sentence "when all players select actions according to the strategy" should be "when all players select actions according to the strategy profile", to be precise.
- Section 2.1.3. Nash Equilibrium:
If we're not going to narrow the focus to 2P0S, then let's not use the terms "exploitability". Instead, call it NashConv or Nash distance (and you can state that this is called "exploitability" in 2P0S games). Regardless, I don't think we should call Equation 2 the "exploitability of player i", since it's about i deviating, not -i.
- Section 3.1.: Again, we have description of \beta and \delta that do NOT match the descriptions given in the Introduction. Now, diversity is the size of an acceptable strategy space, and style is the similarity between a strategy distribution and a preferred strategy. However, later in the section, style is defined as the distance between a single strategy (not distribution) and a preferred strategy.
- Section 3.2.: Conjecture 3.1: "In a 2P0S game, if the set of Nash equilibria forms a convex polyhedron" -- isn't this always true?
- Section 4.1: Equation 13: why does the "otherwise" case have a "-1" in the numerator and the denominator? E.g. if the deltas are all 1, then we get 0/0, which is undefined.
- After Equation 13, maybe "additionally" should say "alternatively"?
- "We prove in the Appendix B.2 that the Pref-CFR algorithm can converge to the NE" -- shouldn't this clarify that this is only in 2P0S? In fact, even in the appendix, I don't see anywhere that clarifies that this is only in 2P0S.
- Equation 16 notation needs work: argmax over a^i, over a function of a, not a^i? There should be an indicator function with equality to a, right?
- Experiments:
"In Heads-Up play, this setup achieved exploitability below 4 mBB/h" -- is this referring to Brown et al., 2018, or to your reproduced results?

**Relation To Broader Scientific Literature:**

The paper’s motivation draws heavily from the poker community, especially research on finding low-exploitability strategies using CFR variants. This is a well-established area with strong ties to multi-agent reinforcement learning and computational game theory. The paper aims to bridge preference-based methods (style/diversity) with standard CFR-like approaches.

**Theoretical Claims:**

I did not review the proofs of the method for correctness.

---

> ### Author Rebuttal · Authors · 2025-04-01
>
> Thank you for your feedback. We first answer your questions and comments in order (due to word limit, the original text is not quoted), and then we summarize some of the Experiments questions and answer them. If there are still questions or new questions arise, please feel free to discuss further.
>
> ---
>
> ## Questions
>
> 1. Indeed, this is a mistake and we will fix it.
> 2. We need to re-write the descriptions of $β$ and $δ$. In this section, I accidentally mistyped diversity and style. Diversity refers to the scale of the acceptable strategy space. Style refers to how close a trained strategy is to another preferred strategy.   $δ$ incentivises a style and $β$ enforces it (at least in poker). Without $β$ (i.e. with a $β$ of 0), if there is only one Nash equilibrium, $δ$ is useless: Pref-CFR will find the original NE.
> 3. Setting only $δ$ indeed means setting $β$ to 0. $ε$ is a property inherent in the strategy, while $β$ is an artificially set parameter. These are two concepts so we use two symbols. We should say $ε \leq β$, and we will fix this problem.
> 4. "ju et al." should be fixed.
> 5. We will clarify that FP converges in 2P0S.
> 6. See (7).
> 7. The goal of this paper is to find a strategy that can take into account people's needs for "style" and "diversity" better than the equilibrium strategies (the NE strategy in two-player zero-sum games and the coarse correlated equilibrium (CCE) strategy in multi-player games). This divides our research into two parts:
>     * How to define the strategies of "style" and "diversity"?
>     * How to find strategies with different styles and diversities during the iteration?
> Therefore, the related work of this article is introduced in two parts. Lines 75 to 99 introduce how previous articles solved the equilibrium (finding the NE in 2p0s games and finding the CCE in multi-player games). The content after line 100 introduces what other studies, besides the traditional NE and CCE, have been conducted by predecessors to attempt to find "better" strategies.
> The writing of the related work in this paper does not present the above content well, and we will appropriately modify the wording.
> 8. “Strategy profile” is indeed more precise.
> 9. In general cases, we should indeed replace “exploitability” with NashConv/NE. Moreover, Equation 2 measures "Player i’s deviation incentive" rather than exploitability.
> 10. See (2).
> 11.  A set of Nash equilibria does not always form a convex polyhedron. For instance, Rock-Paper-Scissors has a unique NE at $[1/3,1/3,1/3]$ (a single point).
> 12. Deltas cannot all be 1, We will emphasize this in the main text. And, for instance, if degrees are [1, 1, 2, 3] we choose actions with probability: [0, 0, ⅓, ⅔].
> 13. We indeed meant “alternatively”.
> 14. To clarify, we reiterate our key proofs:
>     * Appendix B.2 demonstrates that introducing preference parameter $δ$ in CFR preserves Blackwell approachability.
>     * Appendix B.3 proves that the vulnerability parameter $β$ modifies the convergence target set of no-regret algorithms.
>
> Implications:
>
>     - Based on B.2, our algorithm inherits the convergence guarantees of no-regret learning (equivalence to Blackwell approachability), converging to NE in 2p0s games and CCE in multiplayer settings.
>     - B.3 establishes that convergence to the target cone $S_{≤β}$ corresponds to the epsilon-equilibrium class.
>
> We acknowledge the insufficient discussion of these connections in the main text and will:
>
>     - Add explicit signposts in Section 4
>     - Include a new subsection in Appendix B to detail the proof roadmap.
>
> 15. This is a mistake. Thanks for spotting it.
> 16. This metric aligns with both Brown et al.’s results and our experimental findings.
>
>
> ---
> ## More Experiments
>
> For more experiments on smaller games, we have integrated Pref-CFR into OpenSpiel, and calculated the exploitability of 2p Texas Hold'em. These experiments show the good scalability and robustness of our algorithm, and we have redrawn the picture. The results of the experiment can be viewed at: https://drive.google.com/file/d/1F-oNqLm_cVqBNHvDYUf5dtQ_Rk1SdMkH/view?usp=drive_link
>
> In addition, we will add more details to Texas Hold'em training in later versions. We cannot show it to you here due to character limitations. You can check our new experiments in the response of the reviewer CeZM and BR1q.
>
> ---
>
> We deeply appreciate your rigorous review. All revisions will be prominently marked in the final manuscript.

---

### Decision · Program_Chairs · 2025-05-01

**Decision:**

Accept (poster)

**Comment:**

Reviewers agreed that the proposed approach is interesting and important at least to a reasonably large community in AI—the poker community. Reviewers found the theoretical results sound and the experimental results supportive of the conclusion. On the other side, reviewers raised on concerns about the clarity and the generality of the proposed framework beyond the three types of games tested in the paper. Some concerns were addressed during the rebuttal and all reviewers agreed that the paper is acceptable.